# Sample Efficient Reinforcement Learning in Mixed Systems through Augmented Samples and Its Applications to Queueing Networks

**Honghao Wei**
Washington State University
honghao.wei@wsu.edu

**Xin Liu**
ShanghaiTech University
liuxin7@shanghaitech.edu.cn

**Weina Wang**
Carnegie Mellon University
weinaw@cs.cmu.edu

**Lei Ying**
University of Michigan, Ann Arbor
leiying@umich.edu

## Abstract

This paper considers a class of reinforcement learning problems, which involve systems with two types of states: stochastic and pseudo-stochastic. In such systems, stochastic states follow a stochastic transition kernel while the transitions of pseudo-stochastic states are deterministic *given* the stochastic states/transitions. We refer to such systems as mixed systems, which are widely used in various applications, including manufacturing systems, communication networks, and queueing networks. We propose a sample efficient RL method that accelerates learning by generating augmented data samples. The proposed algorithm is data-driven and learns the policy from data samples from both real and augmented samples. This method significantly improves learning by reducing the sample complexity such that the dataset only needs to have sufficient coverage of the stochastic states. We analyze the sample complexity of the proposed method under Fitted Q Iteration (FQI) and demonstrate that the optimality gap decreases as $\tilde{\mathcal{O}}(\sqrt{1/n} + \sqrt{1/m})$, where $n$ is the number of real samples and $m$ is the number of augmented samples per real sample. It is important to note that without augmented samples, the optimality gap is $\tilde{\mathcal{O}}(1)$ due to insufficient data coverage of the pseudo-stochastic states. Our experimental results on multiple queueing network applications confirm that the proposed method indeed significantly accelerates learning in both deep Q-learning and deep policy gradient.

## 1 Introduction

Reinforcement learning (RL) algorithms have recently achieved superhuman performance in gaming, such as AlphaGo (Silver et al., 2017), and AlphaStar (Vinyals et al., 2019), under the premise that vast amounts of training data can be collected. However, collecting data in real-world could be expensive and time-consuming applications such as clinical trials and autonomous driving, posing a significant challenge to extending the success of RL to broader applications. In this paper, we are interested in sample efficient algorithms for a class of problems in which environments (also called systems in this paper) include both stochastic and deterministic transitions, resulting in two types of states: stochastic states and pseudo-stochastic states. For example, in a queueing system, customers arrive and depart, following certain stochastic processes, but the evolution of the queues given arrivals and departures are deterministic. We call these kinds of systems mixed systems. Mixed systems are common in various RL applications, including data centers, ride-sharing systems, and communication networks,

37th Conference on Neural Information Processing Systems (NeurIPS 2023).

which can be modeled as queueing networks so as mixed systems. The broad application of mixed systems makes it crucial to explore whether current RL approaches are efficient in learning optimal policies for such systems. However, existing RL approaches are often not sample efficient due to the curse of dimensionality. For instance, in a queueing system, the state space (i.e. the queue lengths) is unbounded, which means that a tremendous amount of data samples are required to adequately cover the state space in order to learn a near-optimal policy. To highlight the challenge, in a recent paper (Dai and Gluzman, 2022), a 96-core processor with $1,510$ GB of RAM was used to train a queueing policy for a queueing network with only a few nodes.

This paper proposes a new approach to handling the curse of dimensionality based on augmented samples. Our approach is based on two observations:

- When dealing with a mixed system that has a large state space, model-based approaches are incredibly challenging to use. For example, as of today, it remains impossible to analytically determine the optimal policy for a queueing network. On the other hand, using a deep neural network can be regarded as a numerical method for solving a large-scale optimization problem. Therefore, a data-driven, neural network-based solution could be a much more efficient approach.
- It is well-known that training deep learning is data-hungry. While, in principle, model-free approaches can be directly used for mixed systems, they are likely to be very inefficient in sample complexity for mixed systems whose state space is large. In this paper, we will utilize the knowledge of deterministic transitions to generate new data samples by augmenting existing real samples. In other words, we can generalize real samples to a large (even infinite) number of samples that cover the unobserved pseudo-stochastic states. These samples are equally useful for training the neural network as the real samples.

Based on the two observations above, we consider a mixed system that includes two types of states, stochastic states and pseudo-stochastic states, where the transitions of the stochastic states are driven by a stochastic kernel, and the transitions of the pseudo-states are deterministic and conditioned on the current and next stochastic states. We comment that without conditioning the stochastic states, the pseudo-stochastic states become stochastic. The distributions of stochastic states and pseudo-stochastic states are correlated, which makes the problem different from MDPs with exogenous inputs.

With this state separation, we propose an augmented sample generator (ASG). The sample generator generates virtual samples from real ones while keeping stochastic states and augmenting the pseudo-stochastic states. Both the real samples and virtual samples are then used to train the deep neural networks, e.g., Q-networks or policy networks.

We analyze the sample complexity of the proposed approach for mixed systems under Fitted Q Iteration (FQI Ernst et al. (2005)) which is equivalent to DQN (Mnih et al., 2015a) for tabular setting. Specifically, we consider the scenario where the size of pseudo-stochastic state space is much larger than that of stochastic state space, and the set of available real data samples does not provide sufficient coverage of the joint state space. This is the situation where the proposed approach is expected to be particularly advantageous. Our analysis demonstrates that the proposed approach yields a significant improvement in the convergence rate for tabular settings. In particular, by generating $m$ virtual samples for each real sample, the optimality gap between the learned policy and the optimal policy decreases as $\tilde{\mathcal{O}}(\sqrt{1/n} + \sqrt{1/m})$, where $n$ is the number of real samples. Note that without augmented samples, the error is $\tilde{\mathcal{O}}(1)$ due to the lack of data coverage of the pseudo-stochastic states, which reduces to $\tilde{\mathcal{O}}(\sqrt{1/n})$ when we generate at least $n$ augmented samples for each real sample. We also would like to emphasize that $\omega(\sqrt{1/n})$ is also the fundamental lower bound due to the coverage of the stochastic states.

## 1.1 Related Work

Deep reinforcement learning with augmented samples is not new. However, the approach in this paper has fundamental differences compared with existing works in the literature, which we will explain in detail.

**Dyna and its variants:** The first group of related methods is Dyna (Sutton, 1988) and its extensions. Dyna-type algorithms are an architecture that integrates learning and planning for speeding up learning or policy convergence for Q-learning. However, our proposed method differs fundamentally

from these approaches in several aspects: (1) Dyna-type algorithms are model-based methods and need to estimate the system model which will be used to sample transitions. On the contrary, ASG does not require such an estimation and instead leverages the underlying dynamics of the system to generate many augmented samples from one real sample. (2) Dyna is limited to making additional updates only for states that have been observed previously, whereas ASG has the potential to update all pseudo-stochastic states, including those that have not yet been explored. (3) Since Dyna is a model-based approach, the memory complexity is $|\mathcal{S}|^2|\mathcal{X}|^2|\mathcal{A}|$, where $\mathcal{S}$ is the stochastic state space, $\mathcal{X}$ is the pseudo-stochastic state space and $\mathcal{A}$ is the action space. ASG employs a replay buffer only for real data samples, which typically requires far less memory space. Moreover, some Dyna-type approaches like Dyna-$Q^+$ (Sutton, 1988), Linear Dyna (Sutton et al., 2012), and Dyna-$Q(\lambda)$ (Yao et al., 2009), require more computational resources to search the entire state space for updating the order of priorities.

Dyna-type approaches have been successfully used in model-based online reinforcement learning for policy optimization, including several state-of-the-art algorithms, including ME-TRPO (Kurutach et al., 2018), SLBO (Luo et al., 2019), MB-MPO (Clavera et al., 2018), MBPO (Janner et al., 2019), MOPO (Yu et al., 2020). As we already mentioned above, ASG is fundamentally different from these model-based approaches. The key differences between Dyna-type algorithms and ASG are summarized in Table 1.

Table 1: Dyna-type v.s. ASG

| Algorithm | estimate model? | update unseen states? | computational efficient? | store all transitions? | convergence analysis? |
|---|---|---|---|---|---|
| Dyna-type | ✓ | ✗ | ✗ | ✗ | ✗ |
| ASG | ✗ | ✓ | ✓ | ✓ | ✓ |

**Data augmentation:** It has been shown that data augmentation can be used to efficiently train RL algorithms. Laskin et al. (2020) showed that simple data augmentation mechanisms such as cropping, flipping, and rotating can significantly improve the performance of RL algorithms. Fan et al. (2021) used weak and strong image augmentations to learn a robust policy. The images are weakly or strongly distorted to make sure the learned representation is robust. However, these methods rely solely on image augmentation, and none of them consider the particular structure of the mixed system. The purpose of augmentation and distortion is to improve robustness and generalization. Our approach is to use virtual samples, which are as good as real samples to learn an optimal policy. The purpose of introducing the virtual samples is to improve the data coverage for learning the optimal policy.

Encoding symmetries into the designs of neural networks to enforce translation, rotation, and reflection equivariant convolutions of images have also been proposed in deep learning, like G-Convolution (Cohen and Welling, 2016a), Steerable CNN (Cohen and Welling, 2016b), and E(2)-Steerable CNNs (Weiler and Cesa, 2019). Mondal et al. (2020); Wang et al. (2022) investigated the use of Equivariant DQN to train RL agents. van der Pol et al. (2020) imposed symmetries to construct equivariant network layers, i.e., imposing physics into the neural network structures. Lin et al. (2020) used symmetry to generate feasible virtual trajectories for training robotic manipulation to accelerate learning. Our approach relies on the structure of mixed systems to generate virtual samples for learning an optimal policy and can incorporate much more general knowledge than symmetry.

In addition, existing works on low-dimensional state and action representations in RL are also related to our research focus on representing the MDP in a low-dimensional latent space to reduce the complexity of the problem. For example, Modi et al. (2021); Agarwal et al. (2020) studied provably representation learning for low-rank MDPs. Misra et al. (2020); Du et al. (2019) investigated the sample complexity and practical learning in Block MDPs. There are also practical algorithms (Ota et al., 2020; Sekar et al., 2020; Machado et al., 2020; Pathak et al., 2017) with non-linear function approximation from the deep reinforcement learning literature. Although the mixed model studied in this paper can be viewed as an MDP with a low-dimensional stochastic transition kernel, the state space of the MDP does not have a low-dimensional structure, which makes the problem fundamentally different from low-dimensional representations in RL.

**Factored MDP:** A factored MDP (Kearns and Koller, 1999) is an MDP whose reward function and transition kernel exhibit some conditional independence structure. In a factored MDP, the state space is factored into a set of variables or features, and the action space is factored into a set of actions or control variables. This factorization simplifies the representation and allows for more efficient computation and decision-making. While factored MDPs and mixed systems share some similarities, they are actually quite different. In a factored MDP, state variables are grouped into sets, allowing for the factoring of rewards that depend on specific state subsets, and localized (or factorized) transition probabilities. In a mixed system, the cost/reward function is *not* assumed to be factored. Furthermore, although the transition probabilities of stochastic states are localized, depending only on stochastic states and actions, the transitions of pseudo-stochastic states generally depend on the full state vector and are *not* factorized. However, these transitions are *deterministic*, which allows us to generate augmented samples. Given the fundamental differences between the two, the analysis and the results are quite different despite the high-level similarity. Additionally, for the batch offline RL setting without enough coverage used in our paper, there are no existing approaches, including factored MDPs, that guarantee performance in this scenario.

**MDPs with exogenous inputs:** A recent paper considers MDP with exogenous inputs (Sinclair et al., 2022), where the system has endogenous states and exogenous states. The fundamental difference between exogenous-state/endogenous state versus stochastic-state/pseudo-stochastic-state is that the evolution of the exogenous state is independent of the endogenous state, while the evolution of the stochastic state depends on the action. Since the action is chosen based on the pseudo-stochastic state, the evolution of a stochastic state, therefore, depends on the pseudo-stochastic state.

The most significant difference between this paper and existing works is that we propose the concept of mixed systems and mixed system models. Based on the mixed system models, we develop principled approaches to generate virtual data samples, which are as informative for learning as real samples and enhance RL algorithms with much lower sample complexity. Equally significant, we provide sample complexity analysis that theoretically quantifies the sample complexity improvement under the proposed approach and explains the reason that principled data augmentation improves learning in RL.

## 2 Mixed Systems and Mixed System Models

We consider a discrete-time Markov decision process $M = (\mathcal{S} \times \mathcal{X}, \mathcal{A}, P, R, \gamma, \eta_0)$, whose state space is $\mathcal{S} \times \mathcal{X}$, where $\mathcal{S}$ is the set of stochastic states and $\mathcal{X}$ is the set of pseudo-stochastic states, and both $\mathcal{S}$ and $\mathcal{X}$ are assumed to be finite. Further, $\mathcal{A}$ is a finite action space, $R : \mathcal{S} \times \mathcal{X} \times \mathcal{A} \to [0, r_{\max}]$ is a deterministic and known cost function, and $P : \mathcal{S} \times \mathcal{X} \times \mathcal{A} \to \Delta(\mathcal{S} \times \mathcal{X})$ is the transition kernel (where $\Delta(\cdot)$ is the probability simplex).

The transition kernel $P$ is specified in the following way. The transitions of the stochastic state follow a stochastic transition kernel. In particular, the transition of the stochastic state $S_t$ at time $t$ can be represented as

$$p_{ij}(a) = P(S_{t+1} = j | S_t = i, a_t = a). \tag{1}$$

We assume that this transition is independent of the pseudo-stochastic state $X_t$ (and everything else at or before time $t$). The transition of the pseudo-stochastic state is then deterministic, governed by a function $g$ as follows:

$$X_{t+1} = g(S_t, X_t, a_t, S_{t+1}). \tag{2}$$

Our system $M$ is said to be a *mixed system* because it includes both stochastic and deterministic transitions. Our mixed system model then consists of the stochastic transition kernel and the deterministic transition function $g$.

We focus on discounted costs in this paper. Given a mixed system, the value function of a policy $\pi$ is defined as

$$V^\pi(s, x) := \mathbb{E}\left[ \sum_{h=0}^{\infty} \gamma^h R(s_h, x_h, a_h) \middle| (s_0, x_0) = (s, x), a_h = \pi(s_h, x_h) \right], \tag{3}$$

where $\gamma$ is the discount factor, and the expectation is taken over the transitions of stochastic states and randomness in the policy. Let $v^\pi = \mathbb{E}_{(s_0, x_0) \sim \eta_0}[V^\pi(s_0, x_0)]$, i.e., the expected cost when the initial distribution of the state is $\eta_0$. Let $v^* = \min_\pi v^\pi$.

The $Q$-value function of a policy $\pi$ is defined as

$$Q^\pi(s,x,a) := \mathbb{E}\left[\sum_{h=0}^{\infty} \gamma^h R(s_h, x_h, a_h) \middle| (s_0, x_0) = (s,x), a_0 = a, a_h = \pi(s_h, x_h)\right]. \quad (4)$$

Let $Q$ be the optimal $Q$-value function. Then the Bellman equation can be written as

$$Q(s,x,a) = R(s,x,a) + \gamma \mathbb{E}_{s' \sim P(\cdot|s,a)}\left[\min_{a'} Q(s', g(s,x,a,s'), a')\right]. \quad (5)$$

Note that the value function and the $Q$-value function both take values in $[0, V_{\max}]$ for some finite $V_{\max}$ under any policy, due to the assumption that the cost is in $[0, r_{\max}]$.

## 2.1 Example: A Wireless Downlink Network

To better illustrate the structure of mixed systems. Consider the example of a wireless downlink network shown in Fig. 1 with three mobile users. We model it as a discrete-time system such that $\Lambda_t(i)$ is the number of newly arrived data packets for mobile $i$ at the beginning of time slot $t$, and $O_t(i)$ is the number of packets that can be transmitted to mobile $i$ during time slot $t$ if mobile $i$ is scheduled. The base station can transmit to one and only one mobile during each time slot, and let $A_t \in \{1,2,3\}$ denote the mobile scheduled at time slot $t$. The length of queue $i$ evolves as

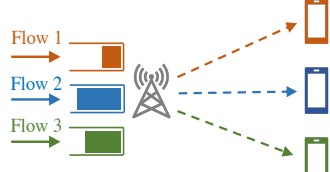

$$q_{t+1}(i) = (q_t(i) + \Lambda_t(i) - O_t(i)\mathbb{I}(A_t = i))^+, \quad (6)$$

where $\mathbb{I}(\cdot)$ is the indicator function and $(\cdot)^+ = \max\{\cdot, 0\}$.

To minimize the total queue lengths, the problem can be formulated as an RL problem such that the state is

Figure 1: A Downlink Wireless Network

$(\mathbf{\Lambda}_t, \mathbf{O}_t, \mathbf{q}_t)$, the action is $A_t$, and the cost is $\sum_i q_t(i)$.
We can see that in this problem, $S_t := (\mathbf{\Lambda}_t, \mathbf{O}_t)$ is the stochastic state and $\mathbf{q}_t$ is the pseudo-stochastic state. In general, the state space of the stochastic states is bounded, e.g., both $\mathbf{\Lambda}_t$ and $\mathbf{O}_t$ may be Bernoulli random variables, but the pseudo-stochastic state such as $\mathbf{q}_t$ is large or even unbounded. Therefore, while the distributions of the stochastic states can be learned with a limited number of samples, learning the distribution of the queue lengths, even under a given policy, will require orders of magnitude more samples. Therefore, vanilla versions of model-free approaches that do not distinguish between stochastic and pseudo-stochastic states require an unnecessary amount of data samples and are not efficient.

Therefore, we propose a sample-efficient, data-driven approach based on deep RL and augmented samples. Augmented samples guarantee enough coverage based on a limited number of real data samples. Note that given $\mathbf{\Lambda}_t$, $\mathbf{O}_t$ and $A_t$, we can generate one-slot queue evolution starting from any queue length.

# 3 Sample Efficient Algorithms for Mixed Systems

## 3.1 Augmented Sample Generator (ASG)

We first introduce the Augmented Sample Generator (ASG, Algorithm 1), which augments a dataset by generating virtual samples. In particular, ASG takes as input a dataset $D$, an integer $m$, and a probability distribution $\beta(\cdot)$ over the pseudo-stochastic states in $\mathcal{X}$. For each sample $(s, x, a, r, s', x') \in D$, we first sample a pseudo-stochastic state $\hat{x}$ from $\beta(\cdot)$, and then construct a virtual sample $(s, \hat{x}, a, \hat{r}, s', \hat{x}')$, where $\hat{r} = R(s, \hat{x}, a)$ and $\hat{x}' = g(s, \hat{x}, a, s')$. Note that we are able to do this since we assume that functions $R$ and $g$ are given in our applications. For each sample in $D$, we repeat this procedure $m$ times independently to generate $m$ virtual samples. Taking the downlink wireless network as an example, where $S_t := (\mathbf{\Lambda_t}, \mathbf{O_t}), X_t := \mathbf{q_t}$, assume that at some timeslot $t$, one of the real samples is

$$(s, x, a, r, s', x') := ((\{3,4,5\}, \{1,2,0\}), \{4,6,6\}, 1, 16, (\{2,3,4\}, \{2,2,0\}), \{6,10,11\}),$$

where the queue length $\mathbf{q_{t+1}}$ is calculated according to Eq. (6). Using ASG, we generate two pseudo-stochastic states (queue length) $\hat{\mathbf{x}}_1 = (1, 2, 3)$, $\hat{\mathbf{x}}_2 = (0, 2, 1)$, then we are able to obtain two virtual samples

$$(s, \hat{x}_1, a, \hat{r}_1, s', \hat{x}_1') := ((\{3, 4, 5\}, \{1, 2, 0\}), \{1, 2, 3\}, 1, 6, (\{2, 3, 4\}, \{2, 2, 0\}), \{3, 6, 8\}),$$

$$(s, \hat{x}_2, a, \hat{r}_2, s', \hat{x}_2') := ((\{3, 4, 5\}, \{1, 2, 0\}), \{0, 2, 1\}, 1, 3, (\{2, 3, 4\}, \{2, 2, 0\}), \{2, 6, 6\}).$$

Note that all the virtual samples represent true transitions of the mixed system if the queue lengths were $\hat{\mathbf{x}}_1$ and $\hat{\mathbf{x}}_2$.

---

**Algorithm 1:** Augmented Sample Generator (ASG)

---

1 **Input**: A dataset $D = \{(s, x, a, r, s', x')\}$, a positive integer $m$, a distribution $\beta(\cdot)$ on $\mathcal{X}$ ;
2 Initialize virtual sample dataset: $D' = \emptyset$ ;
3 **for** *each sample* $(s, x, a, r, s', x') \in D$ **do**
4      Sample $m$ virtual pseudo-stochastic states $\hat{x}_1, \hat{x}_2, \ldots, \hat{x}_m$ from $\beta(\cdot)$ independently;
5      Compute $\hat{r}_i = R(s, \hat{x}_i, a)$ and $\hat{x}_i' = g(s, \hat{x}_i, a, s')$ for $i = 1, 2, \ldots, m$;
6      Add $m$ virtual samples to virtual dataset: $\{(s, \hat{x}_i, a, \hat{r}_i, s', \hat{x}_i'), i = 1, 2, \ldots, m\} \to D'$;
7 **Output:** $D' \cup D$ ;

---

### 3.2 Batch FQI with ASG

We now formally present our algorithm, Batch FQI with ASG (Algorithm 2), for mixed systems. As in a typical setup for batch reinforcement learning (Chen and Jiang, 2019), we assume that we have a dataset $D$ with $n$ samples. The samples are i.i.d. and the distribution of each sample $(s, x, a, r, s', x')$ is specified by a distribution $\mu \in \Delta(\mathcal{S} \times \mathcal{X} \times \mathcal{A})$ for $(s, x, a)$, $r = R(s, x, a)$, $s' \sim P(\cdot|s, a)$, and $x' = g(s, x, a, s')$. The distribution $\mu$ is unknown to the agent.

Our algorithm follows the framework of a batch FQI algorithm. We consider a class of candidate value-functions $\mathcal{F} \in (\mathcal{S} \times \mathcal{X} \times \mathcal{A} \to [0, V_{\max}])$. We focus on the scenario where the number of pseudo-stochastic states, $|\mathcal{X}|$, is far more than that of stochastic states, $|\mathcal{S}|$, and the dataset $D$ does not provide sufficient coverage of the pseudo-stochastic states ($n < |\mathcal{X}|$).

The goal is to compute a near-optimal policy from the given dataset $D$, via finding an $f \in \mathcal{F}$ that approximates the optimal Q-function $Q$. The algorithm runs in episodes. For each episode $k$, it first generates an augmented dataset $D_k$ using $\text{ASG}(D, m, \beta_k)$, where $\beta_k$ is some distribution of the pseudo-stochastic states satisfies that, for each typical pseudo-stochastic state, it can be generated with at least probability $\sigma_1$. We assume that the pseudo-stochastic states will not transition to atypical states, such as extremely large values (possibly infinite) in the queuing example, under a reasonable policy. Details can be found in Assumption A.4 in the supplemental material. We remark that we can also use the same $\beta$ for all the episodes, and in practice, we can simply adopt the estimation of the pseudo-stochastic states (i.e., with Gaussian distribution) as $\beta_k$. The algorithm then computes $f_k$ as the minimizer that minimizes the squared loss regression objective over $\mathcal{F}$. The complete algorithm is given in Algorithm 2.

### 3.3 Guarantee and Analysis

The guarantee of Algorithm 2 is summarized below.

**Theorem 1.** *We assume the data coverage and completeness assumptions (details in the section A in the supplemental material). Given a dataset $D = \{s, x, a, r, s', x'\}$ with $n$ samples, w.p. at least $1 - \delta$, the output policy of Algorithm 2 after $k$ iterations, $\pi_{f_k}$, satisfies*

$$v^{\pi_{f_k}} - v^* \leq \frac{2}{(1-\gamma)^2} \left( \sqrt{\frac{(m+1)C}{m\sigma_1} \left( 5 \left( \frac{1}{n} + \frac{1}{m} \right) V_{\max}^2 \log \left( \frac{nK|\mathcal{F}|^2}{\delta} \right) + \frac{3\delta V_{\max}^2}{n} \right)} \right.$$

$$\left. + \sqrt{c_0 \sigma_2} V_{\max} + \gamma^k (1 - \gamma) V_{\max} \right), \tag{7}$$

*where $C$ is a constant related to the data coverage assumption A.3, and $c_0, \sigma_1, \sigma_2$ are constants defined in Assumption A.4 to ensure a typical set of reasonable typical pseudo-stochastic states.*

---

**Algorithm 2:** Batch FQI with ASG

---

1 **Input**: A dataset $D = \{(s, x, a, r, s', x')\}$, a function class $\mathcal{F}$, a positive integer $m$, distributions $\{\beta_k\}_{k=1}^K$;

2 Initialize approximate $Q$-function: Choose $f_0$ randomly from $\mathcal{F}$.;

3 **for** *episode* $k = 1, 2, \ldots, K$ **do**

4      Generate augmented dataset $\hat{D}_k = \textbf{ASG}(D, m, \beta_k)$;

5      Define loss function as:

$$\mathcal{L}_{\hat{D}_k}(f; f_{k-1}) = \frac{1}{|\hat{D}_k|} \sum_{(s,x,a,r,s',x') \in \hat{D}_k} (f(s,x,a) - r - \gamma V_{f_{k-1}}(s', x'))^2$$

     $f_k = \arg\min_{f \in \mathcal{F}} \mathcal{L}_{\hat{D}_k}(f; f_{k-1}), V_{f_k}(s, x) = \min_{a \in \mathcal{A}} f_k(s, x, a)$;

6      $\pi_{f_k}(s, x) = \arg\min_{a \in \mathcal{A}} f_k(s, x, a)$;

7 **Output:** $\pi_{f_K}$ ;

---

Theorem 1 shows that ASG significantly improves the (real) sample complexity, i.e., the number of real data samples needed. When the tail of pseudo-stochastic states decays fast, e.g. an exponential tails link in a typical queueing network, we can choose a sampling distribution $\beta$ that guarantees that $\sigma_2 = \tilde{\mathcal{O}}(1/n)$ and $\sigma_1 = \Omega(1/\log n)$. Therefore, for sufficiently large $k$ and sufficiently small $\sigma_2$, the first term is the dominating term. As $m$ increases from 1 to $n$, the first term of the bound decreases from $\tilde{\mathcal{O}}(1)$ to $\tilde{\mathcal{O}}(\sqrt{1/n})$. Remark that we cannot establish a convergence result without using ASG (when $m = 0$) since using the dataset $D$ alone doesn't have a sufficient data coverage guarantee, which is critical for batch RL. Due to the page limit, we only present the outline behind our analysis. The detailed proof is deferred to the supplemental material.

**Proof Outline:**

1. First we will show that using performance difference lemma (Kakade and Langford, 2002) it is sufficient by bounding $\|f_k - Q^*\|_{2,\xi}$, where $\xi$ is some distribution.

2. Then, by considering how rare the distribution of the pseudo-stochastic states is under $\xi$ and then classify the pseudo-stochastic states into typical and atypical sets based on a threshold $\sigma_2$ on the distribution. Along with the augmented data generator under $\beta_k(\cdot)$. Then later we show that the term $\|f_k - Q^*\|_{2,\xi}$ can be bounded by $\mathcal{O}\left(\|f_k - \mathcal{T}f_{k-1}\|_{2,\mu_{\beta_k}} + \gamma\|f_{k+1} - Q^*\|_{2,\xi'} + \sqrt{\sigma_2}V_{\max}\right)$, where $\mu_{\beta_k}$ is the data distribution after augmenting virtual samples. The first two terms are related to the typical set given that a sufficient coverage of the data after data augmentation. The last term is easy to obtain by considering the atypical set.

3. Therefore, it is obvious that if we can have a bound on $\|f_k - \mathcal{T}f_{k-1}\|_{2,\mu_{\beta_k}}$ which is independent of the episode $k$, we can prove the final result by expanding $\|f_k - Q^*\|_{2,\xi}$ iteratively for $k$ times. We finally show that the term $\|f_k - \mathcal{T}f_{k-1}\|_{2,\mu_{\beta_k}}$ can indeed be bounded by using the FQI minimizer at each episode and a concentration bound on the offline dataset after augmenting virtual samples.

### 3.4 Extension to the case when $|\mathcal{X}|$ is infinite

Theorem 1 assumes the number of the pseudo-stochastic states is finite. The result can be generalized to infinite pseudo-stochastic state space $|\mathcal{X}|$ if we make the following additional assumption:

**Assumption 1.** *For the typical set $\mathcal{B}$ of pseudo-stochastic states (formally defined in Assumption A.4 in the supplemental material), for any $s, a \in \mathcal{S} \times \mathcal{A}, f \in \mathcal{F}$, if $x \in \mathcal{B}$, then $f(s, x, a) \leq V_{\max}$ otherwise if $x \notin \mathcal{B}$, we have $|f(s, x, a) - Q^*(s, x, a)| \leq V_{\max}$. Furthermore, for any given $f \in \mathcal{F}, (s, x), x \in \mathcal{B}$, we have $|V_f(s', x') - V_f(s'', x'')| \leq V_{\max}$, where $x' = g(s, x, \pi_f, s'), x'' = g(s, x, \pi_f, s'')$.*

The first part in assumption 1 means that the function $f \in \mathcal{F}$ is always bounded when $x \in \mathcal{B}$. Otherwise, the difference between $f$ and the optimal $Q-$function $Q^*$ is bounded, which indicates that although $f$ could be extremely large or even unbounded for the pseudo-stochastic states, including the infinite case, in the atypical set, it is also true for the optimal $Q-$function $Q^*$. Therefore the difference between $f$ and $Q^*$ is assumed to be bounded, which is reasonable in practice. The second

part states that for any given $x \in \mathcal{B}$, the difference between the value functions of any possible next transitions $(s', x'')$ and $(s'', x'')$ is always bounded, which is also true in general especially in queuing literature since the changing of queue lengths between two timeslots is small.

Assumption 1 is easy to be satisfied in a typical queueing system (e.g. the wireless downlink network in Section 2.1). If the control/scheduling policy is a stabilizing policy, then the queues have exponential tails, i.e., the probability decays exponentially as queue length increases, and the policy can drive very large queues to bounded values exponentially fast. Therefore, if all functions in $F$ are from stabilizing policies, Assumption 1 holds because any policy in $F$ including the optimal policy can reduce the queue lengths exponentially fast (in expectation) and the difference in $Q$-functions therefore is bounded.

Under the additional Assumption 1, we show that the same order of the result is achievable. Details can be found in Section A.2 in the supplemental material.

# 4  Experiments

To evaluate the performance of our approach, we compared our results with several baselines in extensive queuing environments. In our simulations, when augmenting virtual samples, we only require that the augmented samples are valid samples for which the action from the real sample remains to be feasible.

## 4.1  The Criss-Cross Network

The criss-cross network is shown in Fig. 2a, which consists of two servers and three classes of jobs. Each job class is queued in a separate buffer if not served immediately. Each server can only serve one job at a time. Server 1 can process jobs of both class 1 and class 3; server 2 can only process jobs of class 2. Class 3 jobs become class 2 jobs after being processed, and class 2 and class 3 jobs leave the system after being processed. The service time of a class $i$ job is exponentially distributed with mean $m_i$. The service rate of a class $i$ job is defined to be $\mu_i := 1/m_i$, and jobs of class 1 and class 3 arrive to the system following the Poisson processes with rates $\lambda_1$ and $\lambda_3$. To make sure the load can be supportable (i.e. queues can be stabilized), we assume $\lambda_1 m_1 + \lambda_3 m_3 < 1$ and $\lambda_1 m_2 < 1$. In the simulation, we consider the imbalanced medium traffic regime, where $\mu_1 = \mu_3 = 2, \mu_2 = 1.5, \lambda_1 = \lambda_3 = 0.6$.

All queues are first-in-first-out (FIFO). Let $e_t(i) \in \{-1, 0, +1\}$ denote the state of class $i$ jobs at time $t$, where $-1$ means a departure, $+1$ means a arrival and $0$ denotes no changes for job $i$. Then $e_t = (e_t(1), e_t(2), e_t(3))$ be the event happens at time $t$ in the system, which is the stochastic state in this mixed system. Let $q_t = (q_t(1), q_t(2), q_t(3))$ be the vector of queue lengths at time $t$, where $q_t(i)$ is the number of class $i$ jobs in the system. Obviously, $q_t$ is the pseudo-stochastic state in the system, which can be derived from $(q_{t-1}, e_{t-1})$ and action $a_t$. We combine ASG with $Q-$learning and compare it with several baselines: (i) the vanilla Q-learning, (ii) a proximal policy optimization (PPO) based algorithm proposed in Dai and Gluzman (2022) designed for queuing networks, (iii) a random policy, and (iv) a priority policy such that the server always serves class 1 jobs when its queue length is not empty, otherwise it serves class 3 jobs. Simulation results in Fig. 2d demonstrate that using ASG significantly improves the performance, and achieves the performance of the optimal policy, which was obtained by solving a large-scale MDP problem, after only 4 training episode ($4k$ training steps). All other policies are far away from the optimal policy, and barely learn anything, as we mentioned that the state space is quite large and is not even sufficiently explored after $4k$ training steps. To further demonstrate the fundamental difference between ASG and Dyna-type algorithms, we also compared ASQ with the model-based approach: Q-learning with Dyna. We can observe that Dyna also fails to learn a good policy since, as we mentioned before, ASG is built for improving sample efficiency with augmented samples that represent *true* transitions for possible all *all* the pseudo-stochastic states, but dyna-type algorithms can only generate addition samples from an estimated model, which are limited by the samples that have been seen before.

## 4.2  Two-Phase Criss-Cross Network:

To further evaluate the power of ASG beyond the tabular setting, We combine ASG with Deep Q-network (Mnih et al., 2015b) named DQN-ASG and evaluate it on a more complicated variant of the

Criss-Cross network presented in Fig. 2b, where class 3 jobs have two phases. Class 3 jobs at phase 1 become phase 2 after being processed with probability $1 - p$, leave the system with probability $p$; and class 3 jobs at phase 2 leave the system after completed processing. Results in Fig. 2e indicate that combining ASG with DQN also has a significant improvement on the performance, and our method only needs 100 episodes to achieve a near-optimal policy. We also remark here that the training time for *DQN-ASG* is three times faster than the baseline (4 hours v.s. 12 hours). In the simulation, all the parameters are set to be the same as those in the previous case, and we use $p = 0.8$. Both the criss-cross and two-phase criss-cross networks are continuous-time Markov chains (CTMCs). We use the standard uniformization (Puterman, 2014; Serfozo, 1979) approach to simulate them using discrete-time Markov chains (DTMCs).

We remark here that the two-phase Criss-Cross network cannot be modeled as MDPs with exogenous inputs. Note that the phase a job is in is a stochastic state, which depends on the action (whether a job is selected for service). Since the scheduling action depends on the queue lengths, the evolution of stochastic states depends on the pseudo-stochastic states (the queue lengths). However, the exogenous input in an MDP with exogenous inputs has to be an observable random process independent of the action. Therefore, the current phase of a job of the phases Criss-Cross network is not an exogenous input and the system is not an MDP with exogenous inputs.

## 4.3 Wireless Networks

In addition to the criss-cross networks, we also evaluate our approach on the downlink network described in Section 2.1 with three mobiles. We let the arrivals are Poisson with rates $\Lambda = \{2, 4, 3\}$, and the channel rates $\mathbf{O}$ to be $\{12, 12, 12\}$. We use the Max-Weight (Tassiulas and Ephremides, 1992) algorithm, a throughput-optimal and heavy-traffic delay optimal algorithm, as the baseline. As shown in Fig. 2f, the learned policy outperforms Max-Weight. To the best of our knowledge, no efficient RL algorithms exist that can outperform Max-Weight.

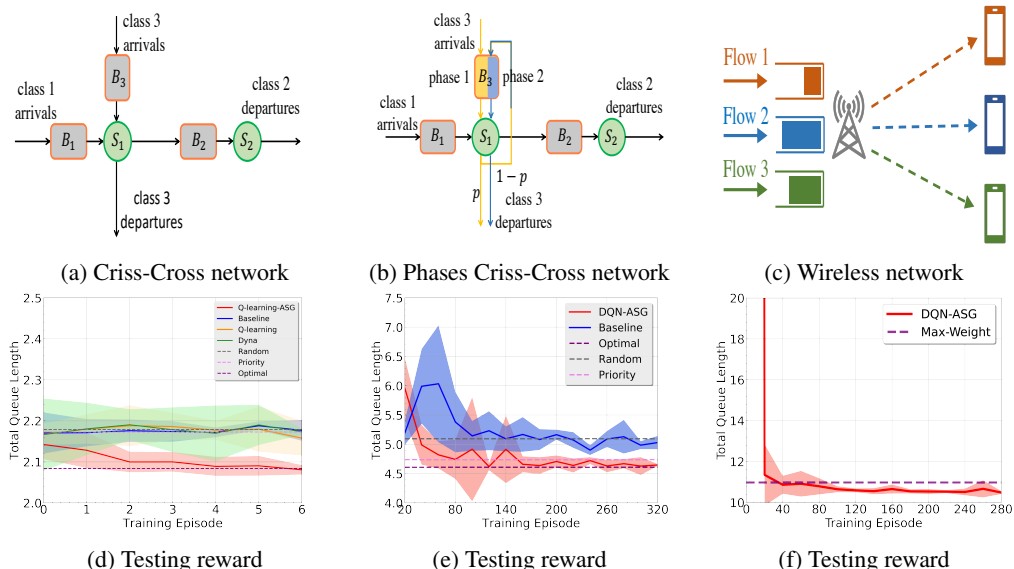

Figure 2: Performance on queuing systems

## 4.4 Additional Simulations on the criss-cross network

To further verify the performance of our algorithm on more scenarios, we first consider a more general criss-cross network where the size of both class 1 and class 3 jobs can vary over time. Then all class 1 and class 3 jobs need multiple timeslots to be processed instead of 1. In particular, when a new job of either class 1 or 2 joins the system, the job size of such job is chosen uniformly. The environment in Fig. 2b can be seen as a special case where class 1, and 3 have a fixed job size 1 and 2, respectively. The performances of different cases are presented in Fig. 3. Our approach still obtains the best result.

The details of the parameters can be found in section B.2. We do not include the optimal solution since, for the general case, the state space is quite large, and using value iteration to obtain the optimal solution is very time-consuming. The priority policy already provides a reasonable good baseline.

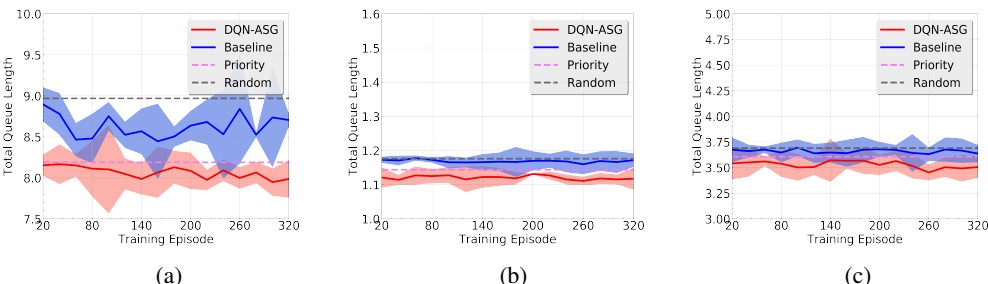

Figure 3: Performance on Criss-Cross network with general job size

## 4.5 Additional Simulations on the Wireless Network

We also evaluate the performance of our algorithm on the wireless network (Fig. 1) under different sets of arrival rates. The results of average queue length compared with those of Max-Weight are summarized in Table 2. We can observe that ASG performs better than Max-Weight in all the cases. Simulation results on a more complicated two-phase wireless network can be found in the supplemental material (Section B.3).

## 4.6 Combining ASG with Policy Gradient-type Algorithm

We also investigate the possibilities of using ASG in policy gradient-type algorithms (i.e., TD3 (Fu et al., 2018), SAC (Haarnoja et al., 2018)) and compare our approach with one of the state-of-art algorithms MBPO (Janner et al., 2019). Our approach still achieves the best performance. Due to page limit, the detailed simulation and algorithm are deferred to Section B in the supplemental material.

| Arrival Rates | Service Rates | Max-Weight | DQN-ASG |
|---|---|---|---|
| $\{2, 3, 4\}$ | $\{12, 12, 12\}$ | 10.979 | **10.403** |
| $\{1, 7, 2\}$ | $\{12, 12, 12\}$ | 13.811 | **13.273** |
| $\{2, 2, 6\}$ | $\{12, 12, 12\}$ | 13.774 | **13.137** |
| $\{3, 1, 5\}$ | $\{12, 12, 12\}$ | 10.895 | **10.091** |

Table 2: Best testing average queue length after training for $120k$ steps

## 5 Conclusions

In this work, we considered RL problems for mixed systems which have two types of states: stochastic states and pseudo-stochastic states. We proposed a sample efficient RL approach that accelerates learning by augmenting virtual data samples. We theoretically quantified the (real) sample complexity gain by reducing the optimality gap from $\tilde{\mathcal{O}}(1)$ to $\tilde{\mathcal{O}}(1/\sqrt{n})$. Experimental studies further confirmed the effectiveness of our approach on several queuing systems. The possible limitation of using ASG is that it introduces more computation during each learning iteration when generating virtual samples, though it is not obvious in our experiments.

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

# Supplementary Material

## A Proof of Theorem 1

In this section, we present the proof of Theorem 1. We first introduce and recall some necessary notations and assumptions. Then, we present some auxiliary lemmas and their proofs. Finally, we combine the lemmas to prove the main result.

**Notations:** Define $P(\nu)$ as distribution over states such that $(s', x') \sim P(\nu) \Leftrightarrow (s, x, a) \sim \nu, s' \sim P(s'|s, a), x' = g(s, a, s')$. In other words, it is the distribution of the next state if the state action pair follows $\nu$. For $f : \mathcal{S} \times \mathcal{X} \times \mathcal{A} \to \mathbb{R}, \nu \in \Delta(\mathcal{S} \times \mathcal{X} \times \mathcal{A})$, where $\Delta(\cdot)$ is the probability simplex, and $p > 1$, define $\|f\|_{p,\nu} = (\mathbb{E}_{(s,x,a)\sim\nu}[|f(s, x, a)|^p])^{1/p}$. Define $\pi_{f,f'}(s, x) := \arg\min_{a\in\mathcal{A}} \min\{f(s, x, a), f'(s, x, a)\}$.

Recall that in Alg. 2, at iteration $k$,

$$\mathcal{L}_{\hat{D}_k}(f; f_{k-1}) = \frac{1}{|\hat{D}_k|} \sum_{(s,x,a,r,s',x')\in\hat{D}_k} (f(s, x, a) - r - \gamma V_{f_{k-1}}(s', x'))^2,$$

where

$$f_k := \arg\min_{f\in\mathcal{F}} \mathcal{L}_{\hat{D}_k}(f; f_{k-1}) \quad \text{and} \quad V_{f_k}(s, x) := \min_{a\in\mathcal{A}} f_k(s, x, a).$$

**Assumptions:**

**Assumption A.1.** *(Realizability) For the optimal policy, $Q^* \in \mathcal{F}$.*

**Assumption A.2.** *(Completeness) For the policy $\pi$ to be evaluated, $\forall f \in \mathcal{F}, \mathcal{T}f \in \mathcal{F}$, where $\mathcal{T} : \mathbb{R}^{\mathcal{S}\times\mathcal{X}\times\mathcal{A}} \to \mathbb{R}^{\mathcal{S}\times\mathcal{X}\times\mathcal{A}}$ is the Bellman update operator, $\forall f$ :*

$$(\mathcal{T}f)(s, x, a) := R(s, x, a) + \gamma\mathbb{E}_{s'\sim P(\cdot|s,a)}[V_f(s', x' = g(s, x, a, s'))].$$

We say a distribution $\nu \in \Delta(\mathcal{S} \times \mathcal{A})$ is admissible in MDP $(\mathcal{S}, \mathcal{X}, \mathcal{A}, P, R, \gamma)$, if there exist $h \geq 0$ and a policy $\pi$ such that $\nu(s, a) = \sum_{x\in\mathcal{X}} \Pr(s_h = s, x_h = x, a_h = a|s_0, x_0, \pi)$. The following assumption is imposed to limit the distribution shift. Note that "admissible" is defined on the stochastic state and action. Later, we will also abuse the notation and call $\nu \in \Delta(\mathcal{S} \times \mathcal{X} \times \mathcal{A})$ is admissible if $\nu(s, a) = \sum_x \nu(s, x, a)$ is admissible.

**Assumption A.3.** *For a data distribution $\mu$, we assume that there exists $C < \infty$ such that for any admissible $v$ and any $(s, a) \in \mathcal{S} \times \mathcal{A}$,*

$$\frac{\nu(s, a)}{\mu(s, a)} \leq C,$$

*where $\nu(s, a) = \sum_{x\in\mathcal{X}} \nu(s, x, a)$ and $\mu(s, a) = \sum_{x\in\mathcal{X}} \mu(s, x, a)$.*

Assumptions A.2-A.3 are standard assumptions in batch reinforcement learning (Chen and Jiang, 2019). However, in assumption A.3, we only require the data coverage of stochastic states which is the fundamental difference.

**Assumption A.4.** *We assume that there exists a set $\mathcal{B}$ of typical pseudo-stochastic states such that the distributions $\beta_k(x), k = 1, \ldots, K$ used for augmenting virtual samples satisfy $\beta_k(x) \geq \sigma_1, \forall x \in \mathcal{B}$. We also assume that the marginal distribution over using a reasonable policy $\pi$, that is, $d_{\eta_0}^\pi(s, x) := (1 - \gamma) \sum_{t=0}^\infty \gamma^{t-1} \Pr(s_t = s, x_t = x|(s_0, x_0) \sim \eta_0, \pi)$ satisfies $\sum_{s\in\mathcal{S}} \sum_{x\in\mathcal{X}, x\notin\mathcal{B}} d_{\eta_0}^\pi(s, x) \leq \sigma_2$, where $\eta_0$ is the initial distribution. Furthermore, if we have for a distribution $\eta$ of the states satisfying $\sum_{s\in\mathcal{S}} \sum_{x\in\mathcal{X}, x\notin\mathcal{B}} \eta(s, x) \leq \sigma_2$, then under any reasonable policy $\pi$, the marginal distribution $\eta_h^\pi(s, x) := \Pr(s_h = s, x_h = x|(s_0, x_0) \sim \eta, \pi)$ satisfies $\sum_{s\in\mathcal{S}} \sum_{x\in\mathcal{X}, x\notin\mathcal{B}} \eta_h^\pi(s, x) \leq c_0\sigma_2, \forall h > 0$, for some constant $c_0 \geq 1$. In particular, all the policies at each iteration, the optimal policy and their joint policies are assumed to be reasonable policies.*

We remark that in a queuing network, under any stable policy, the queue distribution has an exponential tail; in other words, large queue lengths occur with a small probability. In such a case, we can use a uniform distribution for pseudo-stochastic states in set $\mathcal{B}$ to guarantee that $\sigma_1 = \Theta\left(\frac{1}{\log(1/\sigma_2)}\right)$. Therefore, if we choose $\sigma_2 = \frac{1}{n}$, then $\sigma_1 = \frac{1}{\log n}$.

**Auxiliary Lemmas**

In the following lemma, we will show that when all admissible distributions are not far away from the data distribution $\mu$ over stochastic state $\mathcal{S}$ and action $\mathcal{A}$, we can have a good coverage of $\mathcal{S} \times \mathcal{X} \times \mathcal{A}$ by generating virtual samples.

For a given dataset $D$ of size $|D| = n$ and data distribution $\mu$, let $\bar{\mu}_\beta$ denote the expected distribution of the the state action pair $(s, x, a)$ in the combined datase after using Algorithm 2 with a virtual sample distribution $\beta(x)$.

**Lemma A.1.** *Given a virtual sample generating distribution $\beta(x)$ of the pseudo-stochastic state, if $\beta(x) \geq \sigma_1, \forall x \in \mathcal{B}$. Then given any admissible distribution $\nu$, then under Assumptions A.3, we have for any $(s, x, a) \in \mathcal{S} \times \mathcal{X} \times \mathcal{A}, x \in \mathcal{B}$,*

$$\frac{\nu(s, x, a)}{\bar{\mu}_\beta(s, x, a)} \leq \frac{(m+1)C}{m\sigma_1},$$

*where*

$$\bar{\mu}_\beta(s, x, a) = \mu(s, x, a)\frac{n}{nm+n} + \sum_{\hat{x} \in \mathcal{X}} \mu(s, \hat{x}, a)\left(\beta(x)\frac{nm}{nm+n}\right) \tag{8}$$

*Proof.* Given the data distribution $\mu$, we know that the real samples are drawn according to $\mu(s, x, a)$. Then

$$
\begin{aligned}
\frac{\nu(s, x, a)}{\bar{\mu}_\beta(s, x, a)} &\leq \frac{\sum_{\hat{x} \in \mathcal{X}} \nu(s, \hat{x}, a)}{\bar{\mu}_\beta(s, x, a)} = \frac{\sum_{\hat{x} \in \mathcal{X}} \nu(s, \hat{x}, a)}{\sum_{\hat{x} \in \mathcal{X}} \mu(s, \hat{x}, a)} \times \frac{\sum_{\hat{x} \in \mathcal{X}} \mu(s, \hat{x}, a)}{\bar{\mu}_\beta(s, x, a)} \\
&= \frac{\nu(s, a)}{\mu(s, a)} \times \frac{\sum_{\hat{x} \in \mathcal{X}} \mu(s, \hat{x}, a)}{\bar{\mu}_\beta(s, x, a)} \\
&\leq_{(1)} C \times \frac{\mu(s, a)}{\bar{\mu}_\beta(s, x, a)} \\
&=_{(2)} C\left(\frac{\mu(s, a)}{\frac{\mu(s,x,a)}{m+1} + \frac{m}{m+1} \cdot \sum_{\hat{x} \in \mathcal{X}} \mu(s, \hat{x}, a)\beta(x)}\right) \\
&\leq C\left(\frac{(m+1)\mu(s, a)}{m \sum_{\hat{x} \in \mathcal{X}} \mu(s, \hat{x}, a)\beta(x)}\right) \\
&\leq C\left(\frac{(m+1)\mu(s, a)}{m \sum_{\hat{x} \in \mathcal{B}} \mu(s, \hat{x}, a)\beta(x)}\right) \\
&\leq C \cdot \frac{m+1}{m} \cdot \frac{1}{\sigma_1},
\end{aligned}
$$

where the inequality $(1)$ holds because of Assumption A.3, the equality $(2)$ holds by substituting equation (8) and the last inequality is true because the fact that $\beta(x) \geq \sigma_1, \forall x \in \mathcal{B}$. $\qquad\square$

The next lemma transforms the norm in terms of distribution $\nu$ to distribution $\bar{\mu}_\beta$ (Eq. (8)).

**Lemma A.2.** *Let $\nu$ be any admissible distribution, $\bar{\mu}_\beta$ denote the new data distribution defined in Eq. (8) after generating virtual samples with $\beta(x)$. If $\beta(x) \geq \sigma_1, \forall x \in \mathcal{B}$, then under Assumption A.3, for any function $f : \mathcal{S} \times \mathcal{X} \times \mathcal{A} \to \mathbb{R}$, we have $\|f\|_{2,\nu} \leq \sqrt{\frac{m+1}{m}\frac{C}{\sigma_1}}\|f\|_{2,\bar{\mu}_\beta}$, where*

$$\|f\|_{2,\nu} = \left(\sum_{(s,x,a) \in \mathcal{S} \times \mathcal{X} \times \mathcal{A}, x \in \mathcal{B}} |f(s, x, a)|^2 \nu(s, x, a)\right)^{1/2}.$$

*Proof.* For any function $f$, we have

$$\|f\|_{2,\nu} = \left(\sum_{(s,x,a) \in \mathcal{S} \times \mathcal{X} \times \mathcal{A}, x \in \mathcal{B}} |f(s, x, a)|^2 \nu(s, x, a)\right)^{1/2}$$

$$\leq \left( \sum_{(s,x,a)\in\mathcal{S}\times\mathcal{X}\times\mathcal{A}, x\in\mathcal{B}} |f(x,x,a)|^2 \bar{\mu}_\beta(s,x,a) \frac{(m+1)C}{m\sigma_1} \right)^{1/2}$$

$$\leq \sqrt{\frac{(m+1)C}{m\sigma_1}} \|f\|_{2,\bar{\mu}_\beta},$$

where the first inequality is a result of Lemma A.1. $\qquad\square$

**Lemma A.3.** *Consider two functions $f, f' : \mathcal{S}\times\mathcal{X}\times\mathcal{A} \to \mathbb{R}$ and define a policy $\pi_{f,f'}(s,x) := \arg\min_{a\in\mathcal{A}} \min\{f(s,x,a), f'(s,x,a)\}$. Then we have $\forall \nu \in \Delta(\mathcal{S}\times\mathcal{X}\times\mathcal{A})$,*

$$\|V_f - V_{f'}\|_{2,P(\nu)} = \|f - f'\|_{2,P(\nu)\times\pi_{f,f'}}. \tag{9}$$

*Proof.*

$$\|V_f - V_{f'}\|_{2,P(\nu)}^2$$

$$= \sum_{(s,x,a)\in\mathcal{S}\times\mathcal{X}\times\mathcal{A}} \sum_{s'\in\mathcal{S}} P(s'|s,a) \left( \min_{a'\in\mathcal{A}} f(s', g(s,x,a,s'), a') - \min_{a'\in\mathcal{A}} f'(s', g(s,x,a,s'), a') \right)^2$$

$$\leq \sum_{(s,x,a)\in\mathcal{S}\times\mathcal{X}\times\mathcal{A}} \sum_{s'\in\mathcal{S}} P(s'|s,a) \left( f(s', g(s,x,a,s'), \pi_{f,f'}(s', g(s,x,a,s'))) \right.$$

$$\left. - f'(s', g(s,x,a,s'), \pi_{f,f'}(s', g(s,x,a,s'))) \right)^2$$

$$= \|f - f'\|_{2,P(v)\times\pi_{f,f'}}^2.$$

$\qquad\square$

**Lemma A.4.** *Under Assumptions A.3 and A.4, for any admissible distribution $\nu \in \Delta(\mathcal{S}\times\mathcal{X}\times\mathcal{A})$, and a data distribution $\bar{\mu}_\beta$ associated with a virtual sample distribution $\beta(x)$, define $P(\nu)$ as a distribution generated as $s'_i \sim P(\nu)$, then for any policy $\pi$, and $f, f' : \mathcal{S}\times\mathcal{X}\times\mathcal{A} \to \mathbb{R}$, we have*

$$\|f - Q^*\|_{2,\nu} \leq \sqrt{\frac{(m+1)C}{m\sigma}} \|f - \mathcal{T}f'\|_{2,\bar{\mu}_\beta} + \gamma\|f' - Q^*\|_{2,P(\nu)\times\pi_{f',Q^*}} \tag{10}$$

*Proof.*

$$\|f - Q^*\|_{2,\nu} = \|f - \mathcal{T}f' + \mathcal{T}f' - Q^*\|_{2,\nu}$$

$$\leq_{(1)} \|f - \mathcal{T}f'\|_{2,\nu} + \|\mathcal{T}f' - Q^*\|_{2,\nu}$$

$$\leq_{(2)} \sqrt{\frac{(m+1)C}{m\sigma}} \|f - \mathcal{T}f'\|_{2,\bar{\mu}_\beta} + \|\mathcal{T}f' - Q^*\|_{2,\nu}$$

$$\leq_{(3)} \sqrt{\frac{(m+1)C}{m\sigma}} \|f - \mathcal{T}f'\|_{2,\bar{\mu}_\beta} + \gamma\|V_{f'} - V^*\|_{2,P(\nu)}$$

$$\leq \sqrt{\frac{(m+1)C}{m\sigma}} \|f - \mathcal{T}f'\|_{2,\bar{\mu}_\beta} + \gamma\|f' - Q^*\|_{2,P(\nu)\times\pi_{f',Q^*}},$$

where inequality (1) holds because of triangle inequality, inequality (2) comes from lemma A.2, inequality (3) holds because

$$\|\mathcal{T}f' - Q^*\|_{2,\nu}^2 = \|\mathcal{T}^*f' - \mathcal{T}Q^*\|_{2,\nu}^2 = \mathbb{E}_{(s,x,a)\sim\nu}\left[ ((\mathcal{T}f')(s,x,a) - (\mathcal{T}Q^*)(s,x,a))^2 \right]$$

$$= \mathbb{E}_{(s,x,a)\sim\nu}\left[ \left( \gamma\mathbb{E}_{s'\sim P(\cdot|s,a)}\left[ V_{f'}(s', g(s,x,a,s')) - V^*(s', g(s,x,a,s')) \right] \right)^2 \right]$$

$$\leq \gamma^2 \mathbb{E}_{(s,x,a)\sim\nu, s'\sim P(\cdot|s,a)}\left[ (V_{f'}(s', g(s,x,a,s')) - V^*(s', g(s,x,a,s')))^2 \right]$$

$$= \gamma^2 \mathbb{E}_{s'\sim P(\nu)}\left[ (V_{f'}(s', g(s,x,a,s')) - V^*(s', g(s,x,a,s')))^2 \right]$$

$$= \gamma^2 \|V_{f'} - V^*\|_{2,P(\nu)}^2,$$

and the last inequality holds due to Lemma A.3.

$\qquad\square$

**Lemma A.5.** *For a given data sample* $(s, x, a, r, s', a')$ *generated from a data distribution* $\mu$, *such that* $(s, x, a) \sim \mu, s' \sim P(\cdot|s, a), x' = g(s, x, a, s')$, *for any* $f, f' \in \mathcal{F}$, *define* $V_f(s, x) = \min_{a'} f(s, x, a')$, *then*

$$\mathbb{E}\left[(f(s, x, a) - r - \gamma V_{f'}(s', x'))^2\right]$$
$$= \|f - \mathcal{T}f'\|_{2,\mu}^2 + \gamma^2 \mathbb{E}_{(s,x,a) \sim \mu}[\text{Var}(V_{f'}(s', x')|s, x, a)] \tag{11}$$

*Proof.*

$$\mathbb{E}\left[(f(s, x, a) - r - \gamma V_{f'}(s', x'))^2\right]$$
$$= \mathbb{E}\left[(f(s, x, a) - (\mathcal{T}f')(s, x, a) + (\mathcal{T}f')(s, x, a) - (r + \gamma V_{f'}(s', x')))^2\right]$$
$$= \mathbb{E}\left[(f(s, x, a) - (\mathcal{T}f')(s, x, a))^2\right] + \mathbb{E}\left[((\mathcal{T}f')(s, x, a) - (r + \gamma V_{f'}(s', x')))^2\right]$$
$$+ \underbrace{2\mathbb{E}\left[(f(s, x, a) - (\mathcal{T}f')(s, x, a))((\mathcal{T}f')(s, x, a) - (r + \gamma V_{f'}(s', x')))\right]}_{(1)=0}$$
$$= \mathbb{E}\left[(f(s, x, a) - (\mathcal{T}f')(s, x, a))^2\right] + \gamma^2 \mathbb{E}_{(s,x,a) \sim \mu}[\text{Var}(V_{f'}(s', x')|s, x, a)]$$
$$= \|f - \mathcal{T}f'\|_{2,\mu}^2 + \gamma^2 \mathbb{E}_{(s,x,a) \sim \mu}[\text{Var}(V_{f'}(s', x')|s, x, a)],$$

where the equation $(1) = 0$ because that condition on $(s, x, a)$, we have $f$ and $V_{f'}$ are independent. $\square$

**Lemma A.6.** *Under Algorithm 2, at iteration* $k$, *we have*

$$\mathcal{L}_{\hat{\mu}_{\beta_k}}(f; f') - \mathcal{L}_{\hat{\mu}_{\beta_k}}(\mathcal{T}f; f') = \|f - \mathcal{T}f'\|_{2,\bar{\mu}_{\beta_k}}^2, \tag{12}$$

*where* $\mathcal{L}_{\hat{\mu}_{\beta_k}}(f; f') = \mathbb{E}[\mathcal{L}_{\hat{D}_k}(f; f')]$.

*Proof.* Recall that $\hat{D}_k = D \cup D_k$ and $|\hat{D}_k| = nm + n$. The expectation is w.r.t. the random draw of the dataset $D$ and the random generation of dataset $D_k$ with virtual sample distribution $\beta_k$. We know that

$$\mathcal{L}_{\hat{D}_k}(f; f') = \frac{1}{|\hat{D}_k|} \sum_{(s,x,a,r,s',x') \in D} (f(s, x, a) - r - \gamma V_{f'}(s', x'))^2$$
$$+ \frac{1}{|\hat{D}_k|} \sum_{(s,x,a,r,s',x') \in D_k} (f(s, x, a) - r - \gamma V_{f'}(s', x'))^2$$

Let $\mathcal{M}_k^{(s,x,a,s',x')}$ denote the set of virtual samples that are associated with the real sample $(s, x, a, s', x')$ at iteration $k$. Then

$$\mathcal{L}_{\hat{\mu}_{\beta_k}}(f; f') := \mathbb{E}[\mathcal{L}_{\hat{D}_k}(f; f')]$$
$$= \frac{n}{nm + n}\left(\|f - \mathcal{T}f'\|_{2,\mu}^2 + \gamma^2 \mathbb{E}_{(s,x,a) \sim \mu}[\text{Var}(V_{f'}(s', x')|s, x, a)]\right) \quad \text{(by using Lemma } A.5)$$
$$+ \frac{1}{nm + n}\mathbb{E}\left[\sum_{(s,x,a,r,s',x') \in D} \sum_{(s,\bar{x},a,\bar{r},\bar{s}',\bar{x}') \in \mathcal{M}_k^{(s,x,a,r,s',x')}} (f(s, \bar{x}, a) - \bar{r} - \gamma V_{f'}(\bar{s}', \bar{x}'))^2\right]$$
$$= \frac{n}{nm + n}\left(\|f - \mathcal{T}f'\|_{2,\mu}^2 + \gamma^2 \mathbb{E}_{(s,x,a) \sim \mu}[\text{Var}(V_{f'}(s', x')|s, x, a)]\right)$$
$$+ \frac{1}{nm + n}\mathbb{E}\left[\sum_{(s,x,a,r,s',x') \in D} \mathbb{E}\left[\sum_{(s,\bar{x},a,\bar{r},\bar{s}',\bar{x}') \in \mathcal{M}_k^{(s,x,a,r,s',x')}} (f(s, \bar{x}, a) - \bar{r} - \gamma V_{f'}(\bar{s}', \bar{x}'))^2 \Bigg| s, x, a, r, s', x'\right]\right]$$
$$= \frac{n}{nm + n}\left(\|f - \mathcal{T}f'\|_{2,\mu}^2 + \gamma^2 \mathbb{E}_{(s,x,a) \sim \mu}[\text{Var}(V_{f'}(s', x')|s, x, a)]\right)$$

$$+\frac{m}{nm+n}\mathbb{E}\left[\sum_{(s,x,a,r,s',x')\in D}\sum_{\bar{x}\in\mathcal{X}}\beta_k(\bar{x})\left(f(s,\bar{x},a)-\bar{r}-\gamma V_{f'}(\bar{s}',\bar{x}')\right)^2\right]$$

$$=\frac{n}{nm+n}\left(\|f-\mathcal{T}f'\|_{2,\mu}^2+\gamma^2\mathbb{E}_{(s,x,a)\sim\mu}[\mathrm{Var}(V_{f'}(s',x')|s,x,a)]\right)$$

$$+\frac{mn}{nm+n}\sum_{(s,x,a)\in\mathcal{S}\times\mathcal{X}\times\mathcal{A}}\mu(s,x,a)\sum_{\bar{x}\in\mathcal{X}}\beta_k(\bar{x})\left(f(s,\bar{x},a)-\bar{r}-\gamma V_{f'}(\bar{s}',\bar{x}')\right)^2$$

$$=\frac{n}{nm+n}\left(\|f-\mathcal{T}f'\|_{2,\mu}^2+\gamma^2\mathbb{E}_{(s,x,a)\sim\mu}[\mathrm{Var}(V_{f'}(s',x')|s,x,a)]\right)$$

$$+\frac{mn}{nm+n}\sum_{(s,\bar{x},a)\in\mathcal{S}\times\mathcal{X}\times\mathcal{A}}\mu(s,a)\beta_k(\bar{x})\left(f(s,\bar{x},a)-\bar{r}-\gamma V_{f'}(\bar{s}',\bar{x}')\right)^2$$

$$=\frac{n}{nm+n}\left(\|f-\mathcal{T}f'\|_{2,\mu}^2+\gamma^2\mathbb{E}_{(s,x,a)\sim\mu}[\mathrm{Var}(V_{f'}(s',x')|s,x,a)]\right)$$

$$+\frac{nm}{nm+n}\left(\|f-\mathcal{T}f'\|_{2,\mu_k}^2+\gamma^2\mathbb{E}_{(s,x,a)\sim\mu_k}[\mathrm{Var}(V_{f'}(s',x')|s,x,a)]\right),\qquad\text{(by using Lemma }A.5)$$

where $\bar{r}=R(s,\bar{x},a),\mu_k(s,x,a)=\sum_{x'\in\mathcal{X}}\mu(s,x',a)\beta_k(x)=\mu(s,a)\beta_k(x)$. Since we have $\bar{\mu}_{\beta_k}(s,x,a)=\frac{1}{m+1}\mu(s,x,a)+\frac{m}{m+1}\mu(s,a)\beta_k(x)$.

Therefore,

$$\mathcal{L}_{\hat{\mu}_{\beta_k}}(f;f')-\mathcal{L}_{\hat{\mu}_{\beta_k}}(\mathcal{T}f';f')=\|f-\mathcal{T}f'\|_{2,\bar{\mu}_{\beta_k}}^2.$$

$$\square$$

The next lemma shows an upper bound on $\|f_{k+1}-\mathcal{T}f_k\|_{2,\bar{\mu}_{\beta_k}}^2$.

**Lemma A.7.** *Given the MDP $M=(\mathcal{S},\mathcal{X},P,R,\gamma)$, we assume that the $Q-$function classes $\mathcal{F}$ satisfies $\forall f\in\mathcal{F},\mathcal{T}f\in\mathcal{F}$. The dataset $D$ is generated as: $(s,x,a)\sim\mu,r=R(s,x,a),s'\sim P(\cdot|s,a),x'=g(s,x,a,s')$, and the new dataset $\hat{D}_k=D\cup D_k$ is generated by following Alg. 1 with virtual sample generating distribution $\beta_k(x)$ at kth iteration. Then with probability at least $1-\delta,\forall f\in\mathcal{F}$, and $k=0,\dots,K$ we hvae*

$$\|f_{k+1}-\mathcal{T}f_k\|_{2,\bar{\mu}_{\beta_k}}^2\leq 5\left(\frac{1}{n}+\frac{1}{m}\right)V_{\max}^2\log(nK|\mathcal{F}|^2/\delta)+\frac{3\delta V_{\max}^2}{n}\tag{13}$$

*Proof.* Using Lemma A.6 we know that

$$\|f-\mathcal{T}f'\|_{2,\bar{\mu}_{\beta_k}}^2=\mathcal{L}_{\hat{\mu}_{\beta_k}}(f;f')-\mathcal{L}_{\hat{\mu}_{\beta_k}}(\mathcal{T}f;f').$$

Then it is sufficient to bound $\|f-\mathcal{T}f'\|_{2,\bar{\mu}_{\beta_k}}^2$ by bounding

$$\mathcal{L}_{\hat{\mu}_{\beta_k}}(f;f')-\mathcal{L}_{\hat{\mu}_{\beta_k}}(\mathcal{T}f;f')=\mathbb{E}[\mathcal{L}_{\hat{D}_k}(f;f')-\mathcal{L}_{\hat{D}_k}(\mathcal{T}f;f')].$$

For any $f,f'$, recall that

$$\mathcal{L}_{\hat{D}_k}(f;f')=\underbrace{\frac{1}{|\hat{D}_k|}\sum_{(s,x,a,r,s',x')\in D}\left(f(s,x,a)-r-\gamma V_{f'}(s',x')\right)^2}_{\mathcal{L}_D(f;f')}$$

$$+\underbrace{\frac{1}{|\hat{D}_k|}\sum_{(s,x,a,r,s',x')\in D_k}\left(f(s,x,a)-r-\gamma V_{f'}(s',x')\right)^2}_{\mathcal{L}_{D_k}(f;f')}.$$

For any $f,f'$ define

$$Y(f;f'):=(f(s,x,a)-r-\gamma V_{f'}(s',x'))^2-(\mathcal{T}f'(s,x,a)-r-\gamma V_{f'}(s',x'))^2$$

Then for each $(s,x,a,s',x')\in D$, we get i.i.d. variables $Y_1(f;f'),\dots,Y_n(f;f')$.

We also define

$$X_i(f; f') := (f(s_i, \hat{x}_i, a_i) - \hat{r}_i - \gamma V_{f'}(s_i', \hat{x}_i'))^2 - (\mathcal{T}f'(s_i, \hat{x}_i, a_i) - \hat{r}_i - \gamma V_{f'}(s_i', \hat{x}_i'))^2,$$

where $(s_i, \hat{x}_i, a_i, \hat{r}_i, s_i', \hat{x}_i')$ is an augmented sample based on the $i$th real sample $(s_i, x_i, a_i, r_i, s_i')$. Denote the $m$ i.i.d virtual samples by $X_{i_1}(f; f'), \ldots, X_{i_m}(f; f')$. Therefore

$$\mathcal{L}_{\hat{D}_k}(f; f') - \mathcal{L}_{\hat{D}_k}(\mathcal{T}f'; f') = \frac{n}{nm + n} \times \frac{1}{n} \sum_{i=1}^{n} Y_i(f; f') + \frac{nm}{nm + n} \times \frac{1}{nm} \sum_{i=1}^{n} \sum_{j=1}^{m} X_{i_j}(f; f'). \tag{14}$$

Taking the expectations on both sides, we obtain for any $f, f' \in \mathcal{F}$,

$$\mathcal{L}_{\hat{\mu}_{\beta_k}}(f; f') - \mathcal{L}_{\hat{\mu}_{\beta_k}}(\mathcal{T}f'; f') = \frac{n}{nm + n} \mathbb{E}[Y(f; f')] + \frac{nm}{nm + n} \times \frac{1}{n} \mathbb{E}\left[\sum_{i=1}^{n} X_i(f; f')\right]$$

We need to introduce $\frac{1}{n}\sum_{i=1}^{n} Y_i(f; f')$ and $\frac{1}{m}\sum_{i=1}^{n}\sum_{j=1}^{m} X_{i_j}(f; f')$ to bound the above terms. For the first term, we know that the variance of $Y$ can be bounded by:

$$\begin{aligned}
\text{Var}(Y(f; f')) &\leq \mathbb{E}[Y(f; f')^2] \\
&= \mathbb{E}[((f(s, x, a) - r - \gamma V_{f'}(s', x'))^2 - (\mathcal{T}f'(s, x, a) - r - \gamma V_f(s', x'))^2)^2] \\
&= \mathbb{E}\left[(f(s, x, a) - \mathcal{T}f'(s, x, a))^2 (f(s, x, a) + \mathcal{T}f'(s, x, a) - 2r - 2\gamma V_{f'}(s', x'))^2\right] \\
&\leq 4V_{\max}^2 \mathbb{E}\left[(f(s, x, a) - \mathcal{T}f'(s, x, a))^2\right] \\
&= 4V_{\max}^2 \|f - \mathcal{T}f'\|_{2,\mu}^2 \\
&= 4V_{\max}^2 \mathbb{E}[Y(f; f')], \tag{15}
\end{aligned}$$

where the last equality is true because

$$\begin{aligned}
\mathbb{E}[Y(f; f')] &= \mathbb{E}[\mathcal{L}_D(f; f')] - \mathbb{E}[\mathcal{L}_D(\mathcal{T}f'; f')] \\
&= \|f - \mathcal{T}f'\|_{2,\mu}^2 + \gamma^2 \mathbb{E}_{(s,x,a)\sim\mu}[\text{Var}(V_{f'}(s', x')|s, x, a] \\
&\quad - \|\mathcal{T}f' - \mathcal{T}f'\|_{2,\mu}^2 - \gamma^2 \mathbb{E}_{(s,x,a)\sim\mu}[\text{Var}(V_{f'}(s', x')|s, x, a] \qquad \text{(using Lemma A.5)} \\
&= \|f - \mathcal{T}f'\|_{2,\mu}^2.
\end{aligned}$$

Then by applying Bernstein's inequality, together with a union bound over all $f, f' \in \mathcal{F}$, we obtain with probability $1 - \delta$ we have

$$\begin{aligned}
\mathbb{E}[Y(f; f')] - \frac{1}{n}\sum_{i=1}^{n} Y_i(f; f') &\leq \sqrt{\frac{2\text{Var}(Y(f; f'))\log(|\mathcal{F}|^2/\delta)}{n}} + \frac{4V_{\max}^2 \log(|\mathcal{F}|^2/\delta)}{3n} \\
&\leq \sqrt{\frac{8V_{\max}^2 \mathbb{E}[Y(f; f')]\log(|\mathcal{F}|^2/\delta)}{n}} + \frac{4V_{\max}^2 \log(|\mathcal{F}|^2/\delta)}{3n} \tag{16}
\end{aligned}$$

For the second term, note that for any given $i$th sample $(s_i, x_i, a_i, s_i', x_i')$ all the variables $\{X_{i_j}\}$ are i.i.d. Then following a similar argument, then for all $f, f' \in \mathcal{F}$, we have with probability at least $1 - \delta/n$,

$$\mathbb{E}[X_i(f; f')|s_i, x_i, a_i] - \frac{1}{m}\sum_{j=1}^{m} X_{i_j}(f; f')$$

$$\leq \sqrt{\frac{8V_{\max}^2 \mathbb{E}[X_i(f; f')|s_i, x_i, a_i]\log(n|\mathcal{F}|/\delta)}{m}} + \frac{4V_{\max}^2 \log(n|\mathcal{F}|^2/\delta)}{3m}$$

Then it is easy to obtain that we have for all $f, f' \in \mathcal{F}$,

$$\frac{1}{n}\sum_{i=1}^{n} \mathbb{E}\left[X_i(f; f') - \frac{1}{m}\sum_{j=1}^{m} X_{i_j}(f; f')\right]$$

$$\leq \frac{1}{n}\sum_{i=1}^{n}\left(\sqrt{\frac{8V_{\max}^2\mathbb{E}[X_i(f;f')]\log(n|\mathcal{F}|/\delta)}{m}} + \frac{4V_{\max}^2\log(n|\mathcal{F}|^2/\delta)}{3m}\right) + \frac{\delta V_{\max}^2}{n} \quad (17)$$

Combining Eq.(17) and Eq.(16) we can obtain with probability at least $1-\delta$, for all $f,f' \in \mathcal{F}$,

$$\frac{n}{n+nm}\times\mathbb{E}\left[Y(f;f')\right] - \frac{n}{n+nm}\times\frac{1}{n}\sum_{i=1}^{n}Y_i(f;f') + \frac{nm}{nm+n}\times\frac{1}{n}\sum_{i=1}^{n}\mathbb{E}\left[X_i(f;f') - \frac{1}{m}\sum_{j=1}^{m}X_{i_j}(f;f')\right]$$

$$\leq\frac{1}{1+m}\times\left(\sqrt{\frac{8V_{\max}^2\mathbb{E}[Y(f;f')]\log(|\mathcal{F}|^2/\delta)}{n}} + \frac{4V_{\max}^2\log(|\mathcal{F}|^2/\delta)}{3n}\right)$$

$$+ \frac{nm}{nm+n}\left(\frac{1}{n}\sum_{i=1}^{n}\left(\sqrt{\frac{8V_{\max}^2\mathbb{E}[X_i(f;f')]\log(n|\mathcal{F}|/\delta)}{m}} + \frac{4V_{\max}^2\log(n|\mathcal{F}|^2/\delta)}{3m}\right) + \frac{\delta V_{\max}^2}{n}\right)$$
$$(18)$$

Let $f = f_{k+1}, f' = f_k$, then according to Algorithm 2 we know that
$$f_{k+1} = \hat{\mathcal{T}}_{k,\mathcal{F}}f_k := \arg\min_{f\in\mathcal{F}}\mathcal{L}_{\hat{D}_k}(f;f_k).$$

According to Eq. (14), we have

$$\mathcal{L}_{\hat{D}_k}(f;f_k) - \mathcal{L}_{\hat{D}_k}(\mathcal{T}f_k;f_k) = \frac{n}{nm+n}\times\frac{1}{n}\sum_{i=1}^{n}Y_i(f;f_k) + \frac{nm}{nm+n}\times\frac{1}{nm}\sum_{i=1}^{n}\sum_{j=1}^{m}X_{i_j}(f;f_k).$$

Then it is easy to observe that $\hat{\mathcal{T}}_{k,\mathcal{F}}f_k$ minimizes $\mathcal{L}_{\hat{D}_k}(\cdot;f_k)$, it also minimizes

$$\frac{n}{nm+n}\times\frac{1}{n}\sum_{i=1}^{n}Y_i(\cdot;f_k) + \frac{nm}{nm+n}\times\frac{1}{nm}\sum_{i=1}^{n}\sum_{j=1}^{m}X_{i_j}(\cdot;f_k)$$

because the two objectives only differ by a constant $\mathcal{L}_{\hat{D}_k}(\mathcal{T}f_k;f_k)$. Therefore under assumption A.2 we know that $\mathcal{T}f_k \in \mathcal{F}$, we are able to obtain that

$$\frac{1}{nm+n}\sum_{i=1}^{n}Y_i(\hat{\mathcal{T}}_{k,\mathcal{F}}f_k;f_k) + \frac{1}{mn+n}\sum_{i=1}^{n}\sum_{j=1}^{m}X_{i_j}(\hat{\mathcal{T}}_{k,\mathcal{F}}f_k;f_k)$$

$$\leq\frac{1}{nm+n}\sum_{i=1}^{n}Y_i(\mathcal{T}f_k;f_k) + \frac{1}{nm+n}\sum_{i=1}^{n}\sum_{j=1}^{m}X_{i_j}(\mathcal{T}f_k;f_k) = 0, \quad (19)$$

where the last equality holds due to the definitions of $Y_i$ and $X_{i_j}$. Therefore plugging the result from Eq.(19) into Eq.(18), we can obtain

$$\frac{1}{m+1}\mathbb{E}[Y(f_{k+1};f_k)] + \frac{m}{mn+n}\sum_{i=1}^{n}\mathbb{E}[X_i(f_{k+1};f_k)]$$

$$\leq\frac{1}{1+m}\left(\sqrt{\frac{8V_{\max}^2\mathbb{E}[Y(f_{k+1};f_k)]\log(|\mathcal{F}|^2/\delta)}{n}} + \frac{4V_{\max}^2\log(|\mathcal{F}|^2/\delta)}{3n}\right)$$

$$+ \frac{nm}{nm+n}\left(\frac{1}{n}\sum_{i=1}^{n}\left(\sqrt{\frac{8V_{\max}^2\mathbb{E}[X_i(f_{k+1};f_k)]\log(n|\mathcal{F}|^2/\delta)}{m}} + \frac{4V_{\max}^2\log(n|\mathcal{F}|^2/\delta)}{3m}\right) + \frac{\delta V_{\max}^2}{n}\right)$$

By solving the quadratic formula, we get

$$\mathcal{L}_{\hat{\mu}_{\beta_k}}(f_{k+1};f_k) - \mathcal{L}_{\hat{\mu}_{\beta_k}}(\mathcal{T}f_k;f_k) = \|f_{k+1} - \mathcal{T}f_k\|_{2,\bar{\mu}_{\beta_k}}^2$$

$$= \frac{1}{m+1}\mathbb{E}[Y(f_{k+1};f_k)] + \frac{m}{nm+n}\sum_{j=1}^{n}\mathbb{E}[X_i(f_{k+1};f_k)]$$

$$\leq 5\left(\frac{1}{n} + \frac{1}{m}\right)V_{\max}^2\log(n|\mathcal{F}|^2/\delta) + \frac{3\delta V_{\max}^2}{n}$$

Finally, apply a union bound over all $t = 0\dots K$, we conclude the proof.

$\square$

## A.1 Proof of Theorem 1

Now we are ready to show the main theorem. Given a dataset $D$. After generating virtual samples $D_k$ we get a new combined dataset $\hat{D}_k = D \cup D_k$ at each iteration $k$ with virtual sample generating distribution $\beta_k(x)$. We first have

$$
\begin{aligned}
v^{\pi_{f_k}} - v^* =& \frac{1}{1-\gamma} \mathbb{E}_{(s,x) \sim d_{\eta_0}^{\pi_{f_k}}(s,x)} [Q^*(s,x,\pi_{f_k}) - V^*(s,x)] \\
\leq& \frac{1}{1-\gamma} \mathbb{E}_{(s,x) \sim d_{\eta_0}^{\pi_{f_k}}(s,x)} [Q^*(s,x,\pi_{f_k}) - f_k(s,x,\pi_{f_k}) + f_k(s,x,\pi^*) - V^*(s,x)] \\
\leq& \frac{1}{1-\gamma} \left( \|Q^* - f_k\|_{1, d_{\eta_0}^{\pi_{f_k}}(s,x) \times \pi_{f_k}} + \|Q^* - f_k\|_{1, d_{\eta_0}^{\pi_{f_k}}(s,x) \times \pi^*} \right) \\
\leq& \frac{1}{1-\gamma} \left( \|Q^* - f_k\|_{2, d_{\eta_0}^{\pi_{f_k}}(s,x) \times \pi_{f_k}} + \|Q^* - f_k\|_{2, d_{\eta_0}^{\pi_{f_k}}(s,x) \times \pi^*} \right),
\end{aligned}
\tag{20}
$$

where the first equality follows from the performance difference lemma (Kakade and Langford, 2002), the first inequality holds because $\pi_{f_k} \in \arg\min_a f_k(s,x,a)$ and the last inequality is true by using the fact that for any vector $a = (a_1, \ldots, a_n)$ and a valid distribution $d = (d_1, \ldots, d_n), \sum_i d_i = 1$

$$
\|a\|_{1,d} = \sum_i |a_i| d_i = \sum_i |a_i| \sqrt{d_i} \sqrt{d_i} \qquad \text{(Cauchy–Schwarz inequality)}
$$

$$
\leq \sqrt{\sum_i d_i} \times \sqrt{\sum_i |a_i|^2 d_i} = \|a\|_{2,d}.
$$

According to Assumption A.4, we know that $\sum_{s \in \mathcal{S}} \sum_{x \in \mathcal{X}, x \notin \mathcal{B}} d_{\eta_0}^{\pi_{f_k}}(s,x) \leq \sigma_2$, which implies that $\sum_{(s,x,a) \in \mathcal{S} \times \mathcal{X} \times \mathcal{A}, x \notin \mathcal{B}} \{d_{\eta_0}^{\pi_{f_k}}(s,x) \times \pi^*\}(s,x,a) \leq \sigma_2$, and $\sum_{(s,x,a) \in \mathcal{S} \times \mathcal{X} \times \mathcal{A}, x \notin \mathcal{B}} \{d_{\eta_0}^{\pi_{f_k}}(s,x) \times \pi_{f_k}\}(s,x,a) \leq \sigma_2$. Define $\xi = \{d_{\eta_0}^{\pi_{f_k}}(s,x) \times \pi_{f_k}\}(s,x,a)$, then we have

$$
\begin{aligned}
\|f_k - Q^*\|_{2,\xi} =& \left( \sum_{(s,x,a) \in \mathcal{S} \times \mathcal{X} \times \mathcal{A}} |f_k(s,x,a) - Q^*(s,x,a)|^2 \xi(s,x,a) \right)^{1/2} \\
\leq& \left( \sum_{(s,x,a) \in \mathcal{S} \times \mathcal{X} \times \mathcal{A}, x \in \mathcal{B}} |f_k(s,x,a) - Q^*(s,x,a)|^2 \xi(s,x,a) \right)^{1/2} \\
&+ \left( \sum_{(s,x,a) \in \mathcal{S} \times \mathcal{X} \times \mathcal{A}, x \notin \mathcal{B}} |f_k(s,x,a) - Q^*(s,x,a)|^2 \xi(s,x,a) \right)^{1/2} \\
\leq& \|f_k - Q^*\|_{2,\xi} + \sqrt{\sigma_2} V_{\max} \\
\leq& \sqrt{\frac{(m+1)C}{m\sigma_1}} \|f_k - \mathcal{T}f_{k-1}\|_{2,\bar{\mu}_{\beta_k}} + \gamma \|f_{k-1} - Q^*\|_{2,P(\xi) \times \pi_{f_{k-1}},Q^*} + \sqrt{\sigma_2} V_{\max},
\end{aligned}
\tag{21}
$$

where the first inequality holds because $\sqrt{a+b} \leq \sqrt{a} + \sqrt{b} \; (a \geq 0, b \geq 0)$ and the last inequality comes from Lemma A.4.

By using Lemma A.7 we have with at least probability $1 - \delta$

$$
\|f_k - \mathcal{T}f_{k-1}\|_{2,\bar{\mu}_{\beta_k}}^2 \leq \|f_k - \mathcal{T}f_{k-1}\|_{2,\bar{\mu}_{\beta_k}}^2 \leq \epsilon_1
\tag{22}
$$

where $\epsilon_1 = 5 \left( \frac{1}{n} + \frac{1}{m} \right) V_{\max}^2 \log(nK|\mathcal{F}|^2/\delta) + \frac{3\delta V_{\max}^2}{n}$. Therefore, we obtain

$$
\|f_k - Q^*\|_{2,\xi} \leq \gamma \|f_{k-1} - Q^*\|_{2,P(\xi) \times \pi_{f_{k-1}},Q^*} + \sqrt{\frac{(m+1)C\epsilon_1}{m\sigma_1}} + \sqrt{\sigma_2} V_{\max}.
\tag{23}
$$

Now define $\xi' = P(\xi) \times \pi_{f_{k-1}},Q^*$. Then based on Assumption A.4, it is easy to obtain $\sum_{(s,x,a) \in \mathcal{S} \times \mathcal{X} \times \mathcal{A}, x \notin \mathcal{B}} \xi'(s,x,a) \leq c_0 \sigma_2$. Note that the distribution $\xi'' = P(\xi') \times \pi_{f_{k-2}},Q^*$ still satisfies $\sum_{(s,x,a) \in \mathcal{S} \times \mathcal{X} \times \mathcal{A}, x \notin \mathcal{B}} \xi''(s,x,a) \leq c_0 \sigma_2$ according to Assumption A.4, because $\pi_{f_{k-1}}, \pi_{f_{k-2}}$ and $Q^*$ are all assumed to be reasonable policies.

Repeating the expansion above for $k$ times, we have

$$\|f_k - Q^*\|_{2,\xi} \le \frac{1-\gamma^k}{1-\gamma}\left(\sqrt{\frac{(m+1)C\epsilon_1}{m\sigma_1}} + \sqrt{c_0\sigma_2}V_{\max}\right) + \gamma^k V_{\max}.$$

All the above analyses are still applied to the case when $\xi = \{d_{\eta_0}^{\pi_{f_k}}(s,x) \times \pi^*\}$. Therefore, it is straightforward to obtain

$$v^* - v^{\pi_{f_k}} \le \frac{2}{(1-\gamma)^2}\left(\sqrt{\frac{(m+1)C\epsilon_1}{m\sigma_1}} + \sqrt{c_0\sigma_2}V_{\max} + \gamma^k(1-\gamma)V_{\max}\right). \tag{24}$$

Substituting $\epsilon_1$ completes the proof.

## A.2 Extension of Theorem 1

We repeat the assumption for extending our main results to the case when $|\mathcal{X}|$ can be infinite such that $f(s,x,a)$ may not be bounded by $V_{\max}$.

**Repeat of Assumption 1:** For the typical set $\mathcal{B}$ of pseudo-stochastic states (defined in Assumption A.4), for any $s,a \in \mathcal{S} \times \mathcal{A}, f \in \mathcal{F}$, if $x \in \mathcal{B}$, then $f(s,x,a) \le V_{\max}$ otherwise if $x \notin \mathcal{B}$, we have $|f(s,x,a) - Q^*(s,x,a)| \le V_{\max}$. Furthermore, for any given $f \in \mathcal{F}, (s,x), x \in \mathcal{B}$, we have $|V_f(s',x') - V_f(s'',x'')| \le V_{\max}$, where $x' = g(s,x,\pi_f,s'), x'' = g(s,x,\pi_f,s'')$.

There are two places we need to pay attention to: $(1)$ : a bound on $\|f_0 - Q^*\|_{2,P(\xi) \times \pi_{f_0,Q^*}}$, $(2)$ : a bound on the variance of $Y$ as shown in Eq. (15). For the first case, it automatically holds due to assumption 1. For the second term, we first have

$$\text{Var}(Y(f;f')) \le \mathbb{E}[Y(f;f')^2]$$
$$=\mathbb{E}[((f(s,x,a) - r - \gamma V_{f'}(s',x'))^2 - (\mathcal{T}f'(s,x,a) - r - \gamma V_f(s',x'))^2)^2]$$
$$=\mathbb{E}\left[(f(s,x,a) - \mathcal{T}f'(s,x,a))^2 (f(s,x,a) + \mathcal{T}f'(s,x,a) - 2r - 2\gamma V_{f'}(s',x'))^2\right]$$
$$=\mathbb{E}\left[(f(s,x,a) - \mathcal{T}f'(s,x,a))^2 (f(s,x,a) - \mathcal{T}f'(s,x,a) + 2\mathcal{T}f'(s,x,a) - 2r - 2\gamma V_{f'}(s',x'))^2\right].$$

We also know that

$$(f(s,x,a) - \mathcal{T}f'(s,x,a) + 2\mathcal{T}f'(s,x,a) - 2r - 2\gamma V_{f'}(s',x'))^2$$
$$\le 2(f(s,x,a) - \mathcal{T}f'(s,x,a))^2 + 2(2\mathcal{T}f'(s,x,a) - 2r - 2\gamma V_{f'}(s',x'))^2$$
$$=2(f(s,x,a) + Q^*(s,x,a) - Q^*(s,x,a) - \mathcal{T}f'(s,x,a))^2 + 8\gamma^2(\mathbb{E}[V'_{f'}(\hat{s},\hat{x})|s,x,a] - V_{f'}(s',x'))^2$$
$$\le 16V_{\max}^2.$$

Therefore, we have $\text{Var}(Y(f;f')) \le 16V_{\max}^2\mathbb{E}[Y(f;f')]$.

Then we can obtain a similar result of the same order, which only differs for some constant $\tilde{c}$ such that

$$v^* - v^{\pi_{f_k}} \le \frac{2\tilde{c}}{(1-\gamma)^2}\left(\sqrt{\frac{(m+1)C\epsilon_1}{m\sigma_1}} + \sqrt{c_0\sigma_2}V_{\max} + \gamma^k(1-\gamma)V_{\max}\right). \tag{25}$$

# B  Additional Simulations

## B.1  Combining PSG with Policy Gradient-type algorithms

In this section, we investigate the possibilities of using ASG in policy gradient-type algorithms. In particular, we use ASG in the phase of policy evaluation. An algorithm (SAC-ASG) that incorporates ASG into SAC is presented in Alg. 3. We also compare our algorithm SAC-ASG with state-of-art Dyna-type model-based approaches, i.e., MBPO (Janner et al., 2019) on the phases criss-cross network environment (Fig. 2b). The simulation results are shown in Fig.4. We can observe that

---
**Algorithm 3:** SAC-ASG
---
1 **Input**: Critic Networks: $Q_{\theta_1}, Q_{\theta_2}$, Target Critic Networks: $Q_{\theta'_1}, Q_{\theta'_2}$ ;
2        Actor-Network: $A_\phi$, Empty sample reply buffer: $\mathcal{D}$, Learning rate: $\lambda$ ;
3 **for** *each iteration* **do**
4     **for** *each environment interaction* **do**
5        Take action $a_t \sim A_\phi(a_t|s_t, x_t)$, observe next state $(s_{t+1}, x_{t+1})$, and reward $r_t$ ;
6        Store the transition into replay buffer: $\mathcal{D} \leftarrow \mathcal{D} \cup \{s_t, x_t, a_t, r_t, s_{t+1}, x_{t+1}\}$ ;
7     **for** *each training step* **do**
8        Sample mini-batch $d$ of $n$ transitions from replay buffer $\mathcal{D}$;
9        **for** *Each virtual training loop* **do**
10           Obtain virtual dataset $d' := \mathbf{ASG(d,m)}$ ;
11           Combine training dataset $d \cup d' := \{s, x, a, r, s', x'\}$ ;
12           $\tilde{a} \leftarrow A_\phi(s', x'), \quad y \leftarrow r + \gamma(\min_{i=1,2} Q_{\theta'_i}(s', x', \tilde{a}) - \alpha \log(A_\phi(\tilde{a}|s', x'))$ ;
13           $J_Q(\theta_i) = (nm+n)^{-1} \sum (y - Q_{\theta_i}(s, x, a))^2$ for $i \in \{1, 2\}$ ;
14           $\theta_i \leftarrow \theta_i - \lambda \nabla_{\theta_i} J_Q(\theta_i)$ for $i \in \{1, 2\}$     // Update Critic networks
15           $J_\pi(\phi) = (nm+n)^{-1} \sum (\alpha \log A_\phi(a|s) - \min_{i=1,2} Q_{\theta_i}(s, a)))$ ;
16           $\phi \leftarrow \phi - \lambda \nabla_\phi J_\pi(\phi)$       // Update Actor network
17           $\theta'_i \leftarrow \tau\theta_i + (1-\tau)\theta'_i$       // Update target network weights
18 **Output:** Actor Network $A_\phi$ ;
---

the performance of our approach is significantly better than the baselines'. We also would like to emphasize that the training time of our approach is much less than that of MBPO (4 hours v.s. 3 days).

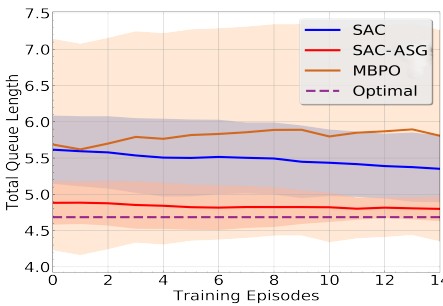

Figure 4: Performance on the Two-phase Criss-Cross Network

## B.2    Details of the Environment in Section 4.4

In this section, we summarize the detailed parameters used in section 4.4 in Table 3.

| Setting | Arrival Rates | Service Rates | Job Size Range |
|---------|---------------|---------------|----------------|
| $(a)$ | $\{0.6, 0.6\}$ | $\{2, 1.5, 1.5\}$ | 2 |
| $(b)$ | $\{0.6, 0.6\}$ | $\{7, 3.5, 7\}$ | 5 |
| $(c)$ | $\{0.6, 0.6\}$ | $\{2.5, 4.5, 2.5\}$ | 5 |

Table 3: Detailed Environment Parameters

## B.3    Two-phase Wireless Network

The two-phase wireless network is a modified version of the downlink network in Section 2.1, which cannot be modeled as an MDP with exogenous inputs. In this example, each packet has two phases. When a packet from a mobile is scheduled, if it is in phase 1, it leaves with probability 0.2 or moves from phase 1 to phase 2; and if it is in phase 2, it leaves the system. We considered Poisson arrivals

with rates $\Lambda = \{2, 4, 3\}$, and the channel rates $O = \{12, 12, 12\}$. As the result shown in Figure. 5 ASG outperforms Max-weight in this environment as well.

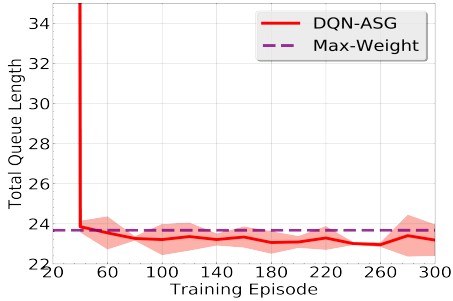

Figure 5: Performance on the Two-phase Downlink Wireless Network

## C   Simulation Details

In our simulations, the reward function is the total queue length (or a scaled version: divided by 20 i the wireless networks). We used Gaussian distributions fitted with real data samples as $\beta_k$, for each real sample, we generated 50 augmented samples and repeated the process 10 times. We use a simple two layer neural network with hidden size 128 with learning rate 0.0003. The batch size used in all the simulations is 256.

### C.1   Experimental settings

For all the simulations, We used a single NVIDIA GeForce RTX 2080 Super with AMD Ryzen 7 3700 8-Core Processor.

