# OpenReview forum: "Sample Efficient Reinforcement Learning in Mixed Systems through Augmented Samples and Its Applications to Queueing Networks"
_NeurIPS.cc/2023/Conference — NeurIPS 2023 spotlight_

### Official Review · Reviewer_ciJb · 2023-06-26

**Soundness:** 3 good
**Presentation:** 3 good
**Contribution:** 4 excellent
**Rating:** 7
**Confidence:** 3

**Summary:**

This paper focuses on a specific class of Markov Decision Processes (MDPs) in which a state can be decomposed into a stochastic component and a pseudo-stochastic component. The transition dynamics of stochastic states depend solely on the current stochastic state and the action taken, while the transition dynamics of pseudo-stochastic states are determined deterministically by the current state, the next stochastic state, and the action. Such MDPs are suitable for modeling applications like queueing systems.

The authors initially consider a scenario with finite state spaces and batch reinforcement learning. Given a set of independent and identically distributed (i.i.d.) transition and reward data, they propose a data augmentation algorithm that generates m new virtual samples from each data point. This is achieved by replacing the pseudo-stochastic components with newly sampled ones. Subsequently, they propose the fitted Q iteration with an augmented dataset. Under certain coverage assumptions for stochastic states and with n initial samples, they establish an error bound on the Q function, which scales as 1/\sqrt{n} + 1/\sqrt{m}. The authors extend their results to scenarios with an infinite pseudo-stochastic state space, along with additional assumptions. Simulations are conducted to demonstrate the benefits of the proposed algorithm.

**Strengths:**

Using reinforcement learning to determine optimal policies for systems with large state spaces is a well-known challenge. However, this paper presents a valuable and intuitive framework that effectively captures the inherent structure of environments, specifically queueing systems, through the introduction of pseudo-stochastic states. This innovative modeling approach is commendable. The concept of utilizing fixed transition functions to generate new data points is novel and has practical relevance for systems influenced by limited stochastic inputs. The theoretical results provided in the paper are sound and contribute to the overall strength of the work.

**Weaknesses:**

In Section 2.1, the authors provide an example where the pseudo-stochastic state space is not finite, which goes beyond the scope of the model as indicated by the assumption of a finite state space in line 148. Although the authors later extend their results, it remains unclear how this example aligns with Assumption 1.

Building upon point 1, Assumption 1 poses challenges in designing the function class F for the Q function. It seems difficult to define a class where the value of an atypical state under any function within the class is consistently close to the optimal Q value. The authors need to provide better justification for why this property can be easily achieved. One approach could be to establish a broad enough function class that satisfies Assumption 1 for a simple queueing system. Justifying the reasonableness of Assumption 1 is crucial, especially considering the paper's claim that the theory and method are applicable to queueing systems with unbounded state spaces.

The simulation section lacks sufficient explanation and some of the plots appear unreasonable. Further details and clarification are needed to address these issues.

There are several typos that need to be corrected throughout the paper.

**Questions:**

In line 93, the authors claim "far less memory space." Could you provide a quantification, such as the complexity mentioned in line 91?

In the simulations, how is the reward function determined? How are the state sampling distributions \beta_k set? Additionally, what is the number of virtual samples used?

Regarding Figures 2 and 3, how is an episode defined in the context of the simulations?

In Figure 3 (a, b, c), it appears that the performance of DQN-ASG does not show improvement after training with episodes. Furthermore, why is the performance in episode 0, where the algorithm should not have learned anything yet, already better than other algorithms? The same question applies to Figure 4 in the appendix.

In lines 24 and 25, from "time-consuming" to the period before "In this paper," the text does not form a complete sentence.

In line 56, the last "and" is unnecessary.

In line 222, please change "satisfies" to "satisfying."

---

> ### Author Rebuttal · Authors · 2023-08-07
>
> We would like to sincerely thank the reviewer for the constructive comments. We will incorporate these great suggestions in our revision and we'd like to address the major concern in the following.
>
> > (1) Building upon point 1, Assumption 1 poses challenges in designing the function class F for the Q function. It seems difficult to define a class where the value of an atypical state under any function within the class is consistently close to the optimal Q value. The authors need to provide better justification for why this property can be easily achieved. One approach could be to establish a broad enough function class that satisfies Assumption 1 for a simple queueing system. Justifying the reasonableness of Assumption 1 is crucial, especially considering the paper's claim that the theory and method are applicable to queueing systems with unbounded state spaces.
>
> - In a typical queueing system, if the control/scheduling policy is a stabilizing policy, then the queues have exponential tails, i.e., the probability decays exponentially as queue length increases, and the policy can drive very large queues to bounded values exponentially fast. Therefore, if all functions in F are from stabilizing policies, Assumption 1 holds because both a policy in F and the optimal policy can reduce the queue lengths exponentially fast (in expectation) and the difference in Q-functions therefore is bounded. The example in Section 2.1 therefore satisfies Assumption 1.  We will add further discussion in the revision.
>
> > (2) In line 93, the authors claim "far less memory space." Could you provide a quantification, such as the complexity mentioned in line 91?
>
> Since our approach is model-free, the complexity of remembering the Q values for the tabular case is $SXA.$
>
> > (3) In the simulations, how is the reward function determined? How are the state sampling distributions $\beta_k$ set? Additionally, what is the number of virtual samples used?
>
> - In our simulations, the reward function is the total queue length (or a scaled version).  We used Gaussian distributions fitted with real data samples as $\beta_k.$  For each real sample, we generated $50$ augmented samples and repeated the process $10$ times. We will include more details about the algorithm in our revised manuscript.
>
> > (4) Regarding Figures 2 and 3, how is an episode defined in the context of the simulations?
>
> Each episode includes 5,000 steps.
>
> > (5)  In Figure 3 (a, b, c), it appears that the performance of DQN-ASG does not show improvement after training with episodes. Furthermore, why is the performance in episode 0, where the algorithm should not have learned anything yet, already better than other algorithms? The same question applies to Figure 4 in the appendix.
>
> We apologize for the confusion. Episode 0 in the label should be episode 20, which is the first testing episode. In our simulations, we test the performance of the learned policy every 20 episodes. DQN-ASG does not improve is because after 20 episodes, DQN-ASG already outperforms the baselines significantly and the performance only improves slowly after that. We include the new figures in the uploaded one-page pdf file.
>
> ---
> We hope the response above addresses the reviewer's concerns. We would also be happy to provide further clarifications if needed. We would also be grateful if the reviewer could consider reevaluating the review and rating based on our response.

---

> > ### Comment · Reviewer_ciJb · 2023-08-11
> >
> > I am satisfied with the authors' response and will consider raising my rating.

---

> > > ### Author Response · Authors · 2023-08-12
> > >
> > > Thank you for your positive feedback and your consideration to raise the rating of our paper!  We're pleased to hear that our response met your satisfaction. Please don't hesitate to let us know if you have any other questions/comments. Thanks!

---

### Official Review · Reviewer_oHp5 · 2023-07-06

**Soundness:** 3 good
**Presentation:** 4 excellent
**Contribution:** 2 fair
**Rating:** 7
**Confidence:** 4

**Summary:**

The paper introduces a novel setting called mixed systems, in which environments comprise both stochastic and deterministic transitions. This leads to two types of states: stochastic states and pseudo-stochastic states. Stochastic states follow a stochastic transition kernel, while pseudo-stochastic states have deterministic transitions given the stochastic states/transitions. Such systems are common in real-world reinforcement learning (RL) applications such as data centers, ride-sharing systems, and communication networks.

Existing RL approaches can suffer from sample inefficiency due to the curse of dimensionality. The paper proposes an augmented sample generator (ASG) to improve sample efficiency. ASG augments transitions by retaining stochastic states while sampling new pseudo-stochastic states.

The paper provides a motivating example of a wireless downlink network, where stochastic states are the number of data packets arriving and the number of packets that can be transmitted at each time step. The pseudo-stochastic state is the length of the queue, which can be computed deterministically given the stochastic states.

The authors offer a convergence guarantee in the form of an optimality gap, connecting the learned value and optimal value with the number of real samples and augmented samples per real sample.

Experiments are conducted on a criss-cross network and the motivating example. The results demonstrate that combining ASG with Q-learning or policy gradient-type algorithms significantly improves performance.

**Strengths:**

1. The paper presents a new setting that is widely observed in real applications.
2. A highly suitable algorithm (ASG) is proposed for this setting.
3. The effectiveness of ASG is demonstrated through solid theoretical proof and several experiments.


**Weaknesses:**

The main weakness of the paper is that the difference between the new mixed systems setting and existing MDPs with exogenous inputs is not entirely clear. Although the analysis in Line 133 is sound, the environments used in this paper can be modeled with MDPs with exogenous inputs. For example, in the wireless downlink network, stochastic state is independent of actions taken. Seems that all environments used can be viewed as MDPs with exogenous inputs, and also a special case of mixed systems. However, the new setting is valuable, and more real-world application examples that can only be modeled with mixed systems are desirable. **If the authors address this issue, I recommend strongly accepting the paper**.

A minor weakness is that the algorithm assumes access to the reward function $R$ and pseudo-stochastic state transition function $g$, potentially limiting its scope. However, this assumption seems reasonable in most examples mentioned.

**Questions:**

1. In Sections 4.1 and 4.2, the system seems to be defined on a continuous time span. It is unclear how continuous time is discretized. I suggest adding details about this issue to the main text or supplementary material.

2. Regarding the wireless downlink network example in Section 2.1, let's consider a situation where the real pseudo-stochastic state has queue lengths of {0, 0, 10}, and a good policy would take action 3 to schedule mobile 3 at this time step. ASG could augment a pseudo-stochastic state as {10, 0, 0}, and in this case, the real action 3 taken seems inappropriate. Although this augmented data still follows the transition function g, it is information less to the policy learning. How will such augmented data affect policy learning and the convergence result?

**Limitations:**

The paper does not extensively discuss limitations or potential negative societal impacts of the work. No further limitations need to be addressed.

---

> ### Author Rebuttal · Authors · 2023-08-07
>
> We first would like to sincerely thank the reviewer for the willingness to reevaluate the rating. We recap your comments and present our detailed response as follows.
>
> > (1) Compared with MDPs with exogenous inputs.
>
> - Our discussion from lines 132-137 explains the fundamental difference. For the environments used in the paper, we would like to point out that the phases Criss-Cross network cannot be modeled as MDPs with exogenous inputs. Note that the phase a job is in is a stochastic state, which depends on the action (whether a job is selected for service). Since the scheduling action depends on the queue lengths,  the evolution of stochastic states depends on the pseudo-stochastic states (the queue lengths). Note that the exogenous input in an MDP with exogenous inputs has to be an observable random process **independent** of the action. Therefore, the current phase of a job of the phases Criss-Cross network is not an exogenous input and the system is not an MDP with exogenous inputs.
>
> - The reviewer is correct that the current wireless network environment can be modeled as an MDP with exogenous inputs. However, if we further consider phase-type service times, then it is not an MDP with exogenous inputs, for a reason similar to the phases Criss-Cross Network. In fact, scheduling in queue systems with nonexponential service times can often be modeled as mixed systems but are not MDPs with exogenous inputs.
>
> To further address the reviewer's question, we considered a two-phase wireless downlink network, where each packet has two phases. When a packet from a mobile is scheduled, if it is in phase 1, it leaves with probability $0.2$ or moves from phase $1$ to phase $2;$ and if it is in phase 2, it leaves the system.
>
> The simulation results are summarized in the following table:
>
> | Episodes| 20 | 60 |100|140|180|220|260|300|
> | ----------- | ----------- | ----------- |  ----------- | ----------- | ----------- | ----------- | ----------- | ----------- |
> | DQN-ASG | 1343.94 | 23.545 | 23.212 | 23.214 |  23.066 |  23.0125  | 23.104 | 22.940
>
> Max-Weight in this setting results in an average total queue length of **23.677**. Our approach still outperforms Max-weight.
>
> Besides queuing systems, we now further include a classical control problem **the Cart Pole problem** as an application of the mixed systems. Details can be found in the one-page pdf file we submitted. Note that the system is not an MDP with exogenous inputs because the stochastic states depend on the pseudo-stochastic states through actions (details in the uploaded PDF).
>
> > (2) In Sections 4.1 and 4.2, the system seems to be defined on a continuous time span. It is unclear how continuous time is discretized. I suggest adding details about this issue to the main text or supplementary material.
>
> - Thanks for the question! They are indeed CTMC. We use the standard uniformization method [R1,R2] to simulate them using DTMCs. We will clarify it in the revision.
>
> > (3)  Regarding the wireless downlink network example in Section 2.1, let's consider a situation where the real pseudo-stochastic state has queue lengths of \{0, 0, 10\}, and a good policy would take action 3 to schedule mobile 3 at this time step. ASG could augment a pseudo-stochastic state as \{10, 0, 0\}, and in this case, the real action 3 taken seems inappropriate. Although this augmented data still follows the transition function g, it is information less to the policy learning. How will such augmented data affect policy learning and the convergence result?
>
> - Thank you for pointing this out. In our simulations, augmented samples should be **valid** samples for which the action from the real sample remains to be feasible. For the example you gave, the augmented sample will be $\{x, y, z\}$ for $z\geq 1,$ e.g. $(10, 0, 1),$ which ensures the action taken for the real sample is a valid action for the augmented sample as well. We will clarify this in the revision.
>
> ---
> We hope that our response addresses your concerns about the applicability and connection to MDPs with exogeous inputs. We are happy to answer any additional questions or concerns. We would also be grateful if the reviewer could consider reevaluating the review and rating based on our response.
>
> ---------------------------references --------------------------------
>
> [R1] Serfozo, Richard F. "An equivalence between continuous and discrete time Markov decision processes." Operations Research 27.3 (1979): 616-620.
>
> [R2] Puterman, Martin L. "Markov decision processes: discrete stochastic dynamic programming." John Wiley \& Sons, 2014.

---

> > ### Author Response · Authors · 2023-08-11
> > **Follow up**
> >
> > Since the discussion period starts, we just wanted to check in and ask if the rebuttal clarified and answered the questions raised in your review. Please don't hesitate to let us know if you have any other questions/comments. We would be very happy to engage further if there are additional questions!

---

> > ### Comment · Reviewer_oHp5 · 2023-08-16
> > **Thanks for your reply.**
> >
> > The authors have provided clear and satisfactory responses to most of my inquiries. As a result, I am inclined to raise my score for this paper. However, a lingering concern remains regarding my second question, specifically pertaining to the wireless downlink network example in Section 2.1. In the example I presented, taking action 3 on the augment pseudo-stochastic state {10, 0, 0} seems technically valid and aligned with the transition functions. Nevertheless, it appears that opting for action 3 could be suboptimal, potentially rendering this augment pseudo-stochastic state ineffective for policy learning. Although this concern seems to have minimal impact given the positive experimental results, I would greatly appreciate it if the authors could address and clarify this question in greater detail, preferably by incorporating it into the main body of the paper.

---

> > > ### Author Response · Authors · 2023-08-17
> > >
> > > Thanks a lot for raising the score of our paper. Much appreciated!  For the follow-up question you have, yes, you are correct that taking action 3 in state {10,0,0} is technically valid. If there is no new packet arrival, action 3 is an idle action and the state remains the same. In the current simulation, we only generate augmented samples {x’>0,y’>0,z’>0} because we know the optimal policy must be work conserving, i.e., it should not schedule an empty queue when some queue is nonempty. Therefore, we restrict our policy space to work conserving policies. In this case, {10, 0, 0} becomes an invalid sample when there is no arrival and is not used in our original simulation.
> > >
> > > However, if we don't restrict our policy space to work-conserving policies, then {10, 0, 0} is always a valid state. In such a case, taking action 3 is suboptimal but is still a useful sample because the agent learns from this sample that action 3 in this state is suboptimal as it incurs a higher cost than action 1 and a worse next state as well.
> > >
> > >  In Q-learning, the agent learns Q-values for all state-action pairs, even for suboptimal actions, so both positive samples and negative samples are useful for learning. To further address your question, we did the simulation for the two-phases wireless network that allows augmented samples to have zero queue length. We provide the performance  below:
> > >
> > > | Episode    | 20 | 60 | 100 |140|180|220|260|300|
> > > | -------- | ------- | ------- | ------- | ------- | ------- | ------- | ------- | ------- |
> > > |ASG-DQN| 105.1667 | 	29.969112 | 	26.64459	 | 25.487535 | 	24.88807	 | 24.109837 | 	23.106613 | 	23.074446 |
> > >
> > > We can observe that our approach still outperforms Max-weight (23.677) but the convergence rate is slightly slower. We believe this is because the state space now is larger as the agent has to consider non-work-conserving policies. We thank the reviewer again for this great question and will add the details including the additional simulation to the revision.
> > >
> > > Do let us know if there is anything else we can address. Always happy to discuss further!

---

### Official Review · Reviewer_d68Z · 2023-07-07

**Soundness:** 4 excellent
**Presentation:** 4 excellent
**Contribution:** 2 fair
**Rating:** 7
**Confidence:** 3

**Summary:**

This paper consider the sample efficiency of RL in a setting involving "mixed systems" where the state space is factored into a stochastic component and another much larger component that is deterministic when conditioned on the stochastic component. The authors develop an algorithm for this setting relying on generating relevant synthetic states that have not been encountered before. This approach when applied to fitted q-learning is able to show sample efficiency bounds for which it is not necessary that the data includes all pseudo-stochastic states. The authors also demonstrate the practical utility of their approach in application to the deep function approximation settings based on queuing networks.

**Strengths:**

- The paper is well written and makes a focused contribution.

- Not only is significant theoretical analysis provided, but also it is shown that this theory makes a meaningful impact for deep function approximation.

- While the empirical domains are not very complex, they do represent real-world use cases, which makes up for this deficiency in my mind.

**Weaknesses:**

- I think the approach will make a big impact where it is applicable, but I worry that this setting is a bit niche. I don't think this is a big problem, but just am a bit less excited as a result.

- It is unclear to what extent the approach can be extended to partially observable domains with more complex input spaces.

- Given the similarities, I would appreciate a more clear connection with the factored MDPs literature.

**Questions:**

1. How realistic is it that the factorization of stochastic states and pseudo-stochastic states is provided to you? Can the cost of identifying these be bounded?

2. Can you talk to the connection between your setting and work on factored MDPs, including existing sample efficiency bounds in that setting? To what degree are you using novel ideas in the bound and approach derivation and to what degree are the insights different from that literature?

3. Do you think that any of the insights of this paper can be useful in partially observable versions of mixed systems, or is it restricted to fully observable settings?



**Limitations:**

The authors do a good job for the most part of discussing assumptions and limitations. I agree that the assumption for the case of an infinite number of pseudo-stochastic states seems mild. Looking at appendix A, the data coverage and completeness assumptions also seem mild, although I think at least more intuition about this should be provided in the main text. One thing I think should be further highlighted and discussed though is that not only do these components of the state space need to exist but they also need to be provided to the user. In my mind, this knowledge about the state space is relatively hard to come by and limits the areas of application. Maybe taking inspiration from the literature on factored MDPs, the authors can account for the cost of the discovery process as well.

---

> ### Author Rebuttal · Authors · 2023-08-07
>
> We appreciate your time and thoughtful evaluation of our paper. We recap your comments and present our detailed response as follows.
> > (1) How realistic is it that the factorization of stochastic states and pseudo-stochastic states is provided to you? Can the cost of identifying these be bounded?
>
> A number of important real-world applications can be modeled as mixed systems. In many of these systems, the separation of stochastic and pseudo-stochastic states is obvious. The paper discussed queueing systems extensively. Here are a few more examples: (i) Water resource management with dynamic water levels where weather and water conditions are stochastic, and the reservoir water level is pseudo-stochastic; and (ii) autonomous driving, where the speed, acceleration, and direction of a vehicle are pseudo-stochastic, and the surrounding environment such as traffic condition and weather of is stochastic.
>
> Of course, there are mixed systems, possibly many, for which it is nontrivial to identify stochastic and pseudo-stochastic states. The cost of separating the states varies from application to application. For these systems, we may take inspiration from the methods for factoring MDPs as suggested by the reviewer, despite factored MDPs and mixed systems have some fundamental differences (see our response to the next question).
>
> This paper focuses on the design of sample-efficient RL algorithms assuming the separation is presented to us, which again is obvious in many applications, including queueing systems.
>
> > (2) Can you talk to the connection between your setting and work on factored MDPs, including existing sample efficiency bounds in that setting? To what degree are you using novel ideas in the bound and approach derivation, and to what degree are the insights different from that literature?
>
> While factored MDPs and mixed systems share some similarities, they are actually quite different. A factored MDP consists of subsets of states, which are weakly correlated. Let  $X_i$ be a state variable and vector $(X_1,\dots,X_n)$ be the full state representation of an MDP. According to the definition in [R1], for a factored MDP, there is a set $Z_i$ associated with each state $i.$ Given $\{Z_i\}\in\{1,\dots,n\},$ the cost (similarly, reward) can be factored as
>
> $
> \begin{equation}
> g(x)=\sum_j g_j(X_{Z_j}),
> \end{equation}
> $
>
> where $X_{Z_j}$ is a subset of the state variables defined by $Z_j$,  and the transition probabilities can be factorized (or localized) as
>
> $P_a(X_i(t+1)\vert X(t) )=P_a(X_i(t+1)\vert X_{Z_i}(t) ),\forall i.$
>
> In a mixed system, the cost/reward function is **not** assumed to be factored. Furthermore, although for the transition probabilities of stochastic states are localized, depending only on stochastic states and actions, the transitions of pseudo-stochastic states **generally** depend on the full state vector and are **not** factorized.
> ```
> For example, in the example of a wireless downlink network in Figure 1, the queue lengths at the next time slot depend on the scheduling decision at the current time slot, which depends on the current values of all the queue lengths, the arrivals, and the channel states.
> ```
>
> However, these transitions are *deterministic*, which allows us to generate augmented samples. Given the fundamental differences between the two, the analysis and the results are quite different despite the high-level similarity. Also, considering the batch off-line RL setting without enough coverage used in our paper, there are no existing approaches, including factored MDP, that guarantee convergence performance in this scenario.
>
> We greatly appreciate the reviewer for bringing up factored MDPs. We will add the comparison to the related work in our revision to clarify the difference.
> > (3) Do you think that any of the insights of this paper can be useful in partially observable versions of mixed systems, or is it restricted to fully observable settings?
>
> Thanks for the question! The insights could be useful for POMDPs. For example, the belief state in POMDPs is typically updated as follows:
>
> $b'(s') = \frac{P(o'\vert s',a)\sum_s P(s'\vert a,s)b(s) }{ \sum_{s'}P(o'\vert s',a)\sum_s P(s'\vert a,s)b(s) }, $
>
> where $s$ denotes the state, $a$ is the action, $o$ stands for the observation state, $P(o|s,a)$ represents the observation probability, and $P(s'|s,a)$ is the transition probability. If the POMDP is also from a mixed system, we may generate augmented samples similarly by sampling according to the brief of stochastic states to improve sample efficiency.
>
> ---
> We hope that our response addresses your concerns about the applicability and connection to factored MDPs and are happy to answer any additional questions or concerns. We would also be grateful if the reviewer could consider reevaluating the review and rating based on our response.
>
> ---------------------------references --------------------------------
>
> [R1] MIT 2.997 Decision-Making in Large-Scale Systems Lecture Note 19, https://ocw.mit.edu/courses/2-997-decision-making-in-large-scale-systems-spring-2004/c578943cb9b5ba63e3976e01de52c61e_lec_19_v1.pdf

---

> > ### Comment · Reviewer_d68Z · 2023-08-15
> > **Response to Rebuttal**
> >
> > Thank you for your response to my questions. I really appreciate your clarification regarding the connection to factored MDPs and POMDPs. I have increased my score as this cleared up some confusion of mine.

---

> > > ### Author Response · Authors · 2023-08-15
> > >
> > > Thank you for your acknowledgement and raising the score of our paper. Much appreciated!

---

### Author Rebuttal · Authors · 2023-08-07

We appreciate the reviewers' insightful comments and suggestions. Your feedback has significantly contributed to enhancing the quality of our work. We would now like to address each reviewer's questions individually to ensure we provide comprehensive responses to your specific concerns. We include two additional simulations in the uploaded pdf. The first addresses comments by Reviewer *oHp5*, particularly concerning distinguishing mixed systems from MDPs with exogenous inputs. The second simulation explores the potential of our approach in other applications.

---

### Decision · Program_Chairs · 2023-09-21

**Decision:**

Accept (spotlight)

**Comment:**

The paper studies sample efficiency of RL in "mixed systems". The theoretical results and simulations are much appreciated and presented in a clean and well-organized manner. The proposed algorithm can make a big impact to queueing systems including wireless communication, resource management, autonomous driving, etc.

There are some initial concerns on the applicability of the mixed systems in practice. Through the author-reviewer discussions, these concerns are largely lifted. In addition, authors provide extended experimental results and explanations to core concept in the paper, such as the difference between the mixed systems and existing MDPs with exogenous inputs. I would encourage authors to carefully incorporate these discussions to the revision of the paper.